# GLOBAL CONVERGENCE OF POLICY GRADIENT IN AVERAGE REWARD MDPS

**Navdeep Kumar**[*]
Electrical and Computer Engineering
Technion - Israel Institute of Technology
navdeepkumar@campus.technion.ac.il

**Yashaswini Murthy**[*]
ECE & CSL
University of Illinois Urbana-Champaign
ymurthy2@illinois.edu

**Itai Shufaro**
Electrical and Computer Engineering
Technion - Israel Institute of Technology
itai.shufaro@campus.technion.ac.il

**Kfir Y. Levy**
Electrical and Computer Engineering
Technion - Israel Institute of Technology
kfirylevy@technion.ac.il

**R. Srikant**
ECE & CSL
University of Illinois Urbana-Champaign
rsrikant@illinois.edu

**Shie Mannor**
Electrical Engineering
Technion - Israel Institute of Technology
NVIDIA Research
shie@ee.technion.ac.il

## ABSTRACT

We present the first comprehensive finite-time global convergence analysis of policy gradient for infinite horizon average reward Markov decision processes (MDPs). Specifically, we focus on ergodic tabular MDPs with finite state and action spaces. Our analysis shows that the policy gradient iterates converge to the optimal policy at a sublinear rate of $O\left(\frac{1}{T}\right)$, where $T$ represents the number of iterations. Performance bounds for discounted reward MDPs cannot be easily extended to average reward MDPs as the bounds grow proportional to the fifth power of the effective horizon. Recent work on such extensions makes a smoothness assumption that has not been verified. Thus, our primary contribution is in providing the first complete proof that the policy gradient algorithm converges globally for average-reward MDPs, without such an assumption. We also obtain the corresponding finite-time performance guarantees. In contrast to the existing discounted reward performance bounds, our performance bounds have an explicit dependence on constants that capture the complexity of the underlying MDP. Motivated by this observation, we reexamine and improve the existing performance bounds for discounted reward MDPs. We also present simulations that empirically validate the result.

## 1 INTRODUCTION

Average reward Markov Decision Processes (MDPs) find applications in domains where decisions are made over time to optimize long-term performance. Some of these applications include resource allocation, portfolio management in finance, healthcare, and robotics (Ghalme et al., 2021; Bielecki et al., 1999; Patrick & Begen, 2011; Mahadevan, 1996; Tadepalli & Ok, 1998). Approaches for determining the optimal policy can be broadly categorized into dynamic programming algorithms (such as value and policy iteration (Murthy et al., 2024; Abbasi-Yadkori et al., 2019; Gosavi, 2004)) and gradient-based algorithms. Although gradient-based algorithms are heavily used in practice (Schulman et al., 2015; Baxter & Bartlett, 2000), the theoretical analysis of their global convergence is a relatively recent undertaking.

---

[*]Equal contribution

While extensive research has been conducted on the global convergence of policy gradient methods in the context of discounted reward MDPs (Agarwal et al., 2020; Khodadadian et al., 2021), comparatively less attention has been given to its average reward counterpart. Contrary to average reward MDPs, the presence of a discount factor ($\gamma < 1$) serves as a source of contraction that alleviates the technical challenges involved in analyzing the performance of various algorithms in the context of discounted reward MDPs. Consequently, many algorithms designed for average reward MDPs are evaluated using the framework of discounted MDPs, where the discount factor approaches one (Grand-Clément & Petrik, 2024).

In the context of discounted reward MDPs, the projected policy gradient (PPG) algorithm converges as

$$\rho_\gamma^* - \rho_\gamma^{\pi_k} \leq O\left(\frac{1}{(1-\gamma)^5}\right) \tag{1}$$

where $\gamma$ is the discount factor, $\pi_k$ is the policy obtained at the $k$-th iteration of the PPG algorithm, $\rho^*\gamma$ represents the optimal value function, and $\rho\gamma^{\pi_k}$ represents the value function iterates obtained through projected gradient ascent (Xiao, 2022b). Let $\rho^\pi$ denote the average reward associated with some policy $\pi$. It is well known that $\rho^\pi = \lim_{\gamma \to 1}(1-\gamma)\rho_\gamma^\pi$ under some mild conditions (Puterman, 1994; Bertsekas, 2007). Utilizing this relationship in equation 1, we observe the upper bound tends to infinity in the limit $\gamma \to 1$. Hence, it is necessary to devise an alternate approach to study the convergence of policy gradient in the context of average reward MDPs.

## 1.1 RELATED WORK

There is a wealth of literature on discounted reward MDPs. Fazel et al. (2018) were among the first to establish the global convergence of policy gradients, specifically within the domain of linear quadratic regulators. Bhandari & Russo (2024) established a connection between the policy gradient and policy iteration objectives, determining conditions under which policy gradient algorithms converge to the globally optimal solution. Agarwal et al. (2020) offer convergence bounds of $O(\frac{1}{\epsilon^2(1-\gamma)^6})$ for policy gradient and $O(\frac{1}{\epsilon(1-\gamma)^2})$ for natural policy gradient, where $\epsilon$ represents the suboptimality. It is noteworthy that while their convergence bounds for policy gradient rely on the cardinality of the state and action space, the convergence bounds for natural policy gradient are independent of them. Xiao (2022b) enhance the $O(\frac{1}{\epsilon^2(1-\gamma)^6})$ policy gradient bounds by refining the dependency on the discount factor, yielding improved bounds of $O(\frac{1}{\epsilon(1-\gamma)^5})$. Zhang et al. (2021) prove that variance-reduced versions of stochastic policy gradient also converge to the global optimal solution. They achieve this through a gradient truncation mechanism. Mei et al. (2020) analyze global convergence of softmax-based gradient methods and prove exponential rejection of suboptimal policies.

The global convergence of policy gradient methods has been extensively studied in the context of planning for average reward Markov Decision Processes (MDPs). Even-Dar et al. (2009) and Murthy & Srikant (2023) provide foundational results for natural policy gradient methods, proving their global convergence for finite state and action spaces. Extending this work, Grosof et al. (2024) analyze the more challenging case of infinite state spaces, establishing theoretical guarantees for convergence in this general setting. A critical contribution of our work lies in proving that the smoothness of the average reward holds for a large class of MDPs in the tabular setting. This smoothness property allows us to establish the first global convergence bounds for policy gradient methods in average reward MDPs, eliminating the need for previously unverified and restrictive smoothness assumptions.

In the learning context, actor-critic and gradient-based methods have been a central focus for analyzing average reward MDPs. Konda & Tsitsiklis (1999) investigate two-time-scale actor-critic algorithms, employing linear function approximation for value functions and demonstrating asymptotic local convergence to a stationary point. Similarly, Bhatnagar et al. (2009) analyze four gradient-based methods, including natural policy gradients, and establish asymptotic local convergence using an ODE-based framework. Addressing global convergence, Bai et al. (2023) study policy gradient methods under the assumption that the average reward is smooth with respect to the policy parameters, achieving a regret bound of $O(T^{1/4})$. Ganesh et al. (2024) consider the average reward natural actor-critic algorithm which does not rely on knowledge of mixing time, and provide a regret bound of $O\sqrt{T}$. However, the results in both of these works rely on unverified smoothness assumptions, leaving

open questions about their applicability to general MDPs. Our contributions further strengthen the understanding of when these assumptions might indeed be true.

## 1.2 CONTRIBUTIONS

In this subsection, we outline the key contributions of this paper.

- **Elimination of Smoothness Assumption:** Unlike previous work which assumes the underlying average cost function is smooth, we prove its smoothness by introducing a new analysis technique. This technique addresses the key difficulty of the lack of uniqueness of the value function in average-reward problems. We overcome this challenge by using a projection technique to ensure uniqueness and leveraging the properties of the projection to prove smoothness. This removes a significant assumption and strengthens the theoretical foundations of policy gradient methods in average reward MDPs.

- **Expression for Smooth Average Cost:** We derive an explicit expression for the average cost that is shown to be smooth in the policy $\pi$. This contribution is critical as it provides a deeper understanding and new insights into the behaviour of the average reward MDPs under policy gradient methods.

- **Sublinear Convergence Bounds:** Using the above smoothness property, we present finite time bounds on the optimality gap over time, showing that the iterates approach the optimal policy with an overall regret of $O\left(\log\left(T\right)\right)$. In contrast, the regret bounds in Bai et al. (2023) are atleast $O\left(T^{\frac{1}{4}}\right)$ without learning error and with tabular parametrization. In place of the discount factor and the cardinality of the state and action spaces in the discounted setting, our finite-time performance bounds involves a different parameter which characterizes the complexity of the underlying MDP.

- **Extension to Discounted Reward MDPs:** Our analysis can also be applied to the discounted reward MDP problem to provide stronger results than the state of the art. In particular, we show that our performance bounds for discounted MDPs can be expressed in terms of a problem complexity parameter, which can be independent of the size of the state and action spaces in some problems.

- **Experimental Validation:** We simulate the performance of policy gradient across a simple class of MDPs to empirically evaluate its performance. The simulations illustrate the impact of MDP complexity on convergence rates. Unlike previous results, where the bounds depend solely on the size of the state and action spaces, these simulations demonstrate how the underlying structure of the MDP can result in significantly different convergence rates, even with fixed state and action spaces. These observations further validate the theoretical bounds derived for the convergence of projected policy gradient in average reward MDPs.

## 2 PRELIMINARIES

In this section, we introduce our model, address the limitations of applying the optimality gap bounds from discounted reward MDPs to the average reward scenario, present the gradient ascent update, and discuss the assumptions underlying our analysis.

### 2.1 AVERAGE REWARD MDP FORMULATION

We consider the class of infinite horizon average reward MDPs with finite state space $\mathcal{S}$ and finite action space $\mathcal{A}$. The environment is modeled as a probability transition kernel denoted by $\mathbb{P}$. We consider a class of randomized policies $\Pi = \{\pi : \mathcal{S} \to \Delta(\mathcal{A})\}$, where a policy $\pi$ maps each state to a probability vector over the action space. The transition kernel corresponding to a policy $\pi$ is represented by $\mathbb{P}^\pi : \mathcal{S} \to \mathcal{S}$, where $\mathbb{P}^\pi(s'|s) = \sum_{a \in \mathcal{A}} \pi(a|s)\mathbb{P}(s'|s,a)$ denotes the single step probability of moving from state $s$ to $s'$ under policy $\pi$. Let $r(s,a)$ denote the single step reward obtained by taking action $a \in \mathcal{A}$ in state $s \in \mathcal{S}$. The single-step reward associated with a policy $\pi$ at state $s \in \mathcal{S}$ is defined as $r^\pi(s) = \sum_{a \in \mathcal{A}} \pi(a|s)r(s,a)$.

The infinite horizon average reward objective $\rho^\pi$ associated with a policy $\pi$ is defined as:

$$\rho^\pi = \lim_{N \to \infty} \frac{\mathbb{E}_\pi \left[ \sum_{n=0}^{N-1} r^\pi(s_n) \right]}{N}, \tag{2}$$

where the expectation is taken with respect to $\mathbb{P}^\pi$. The average reward is independent of the initial state distribution under some mild conditions (Ross, 1983; Bertsekas, 2007) and can be alternatively expressed as $\rho^\pi = \sum_{s \in \mathcal{S}} d^\pi(s) r^\pi(s)$, where $d^\pi(s)$ is the stationary measure corresponding to state $s$ under the transition kernel $\mathbb{P}^\pi$, ensuring that $d^\pi$ satisfies the equation $d^\pi \mathbb{P}^\pi = d^\pi$. Associated with a policy is a relative state value function $v^\pi \in \mathbb{R}^{|\mathcal{S}|}$ that satisfies the following average reward Bellman equation

$$\rho^\pi \mathbb{1} + v^\pi = r^\pi + \mathbb{P}^\pi v^\pi, \tag{3}$$

where $\mathbb{1}$ is the all ones vector (Puterman, 1994; Bertsekas, 2007). Note that $v^\pi$ is unique up to an additive constant. Setting $\sum_{s \in \mathcal{S}} d^\pi(s) v^\pi(s) = 0$ imposes an additional constraint over $v^\pi$, providing a unique value function vector denoted by $v_0^\pi$, known as the basic differential reward function (Tsitsiklis & Van Roy, 1999). It can be shown that $v_0^\pi$ can alternatively expressed as $v_0^\pi(s) = \mathbb{E}_\pi \left[ \sum_{n=0}^{\infty} (r^\pi(s_n) - \rho^\pi) | s_0 = s \right]$. Hence any element in the set $\{v_0^\pi + c\mathbb{1} : c \in \mathbb{R}\}$ is a solution $v^\pi$ to the Bellman equation equation 3.

The relative state action value function $Q^\pi \in \mathbb{R}^{\mathcal{S} \times \mathcal{A}}$ associated with a policy $\pi$ is defined as:

$$Q^\pi(s,a) = r(s,a) + \sum_{\substack{s' \in \mathcal{S} \\ a' \in \mathcal{A}}} \mathbb{P}(s'|s,a)\,\pi(a'|s')Q^\pi(s',a') - \rho^\pi \qquad \forall (s,a) \in \mathcal{S} \times \mathcal{A} \tag{4}$$

Similar to $v^\pi$, $Q^\pi$ is also unique up to an additive constant. Analogously, every solution $Q^\pi$ of equation 4 can be expressed as an element in the set $\{Q_0^\pi(s,a) + c\mathbb{1} : c \in \mathbb{R}\}$ where $Q_0^\pi(s,a) = \mathbb{E}_\pi \left[ \sum_{n=0}^{\infty} (r^\pi(s_n) - \rho^\pi) | s_0 = s, a_0 = a \right]$. Upon averaging equation 4 with policy $\pi$, it follows that $v^\pi(s) = \sum_{a \in \mathcal{A}} \pi(a|s) Q^\pi(s,a)$. The average reward policy gradient theorem (Sutton & Barto, 2018) for policies parameterized by $\theta$ is given by:

$$\frac{\partial \rho}{\partial \theta} = \sum_{s \in \mathcal{S}} d^\pi(s) \sum_{a \in \mathcal{A}} \frac{\partial \pi(s,a)}{\partial \theta} Q^\pi(s,a) \tag{5}$$

As we focus on tabular policies in this paper, our parameterization aligns with the tabular policy, where $\theta$ is equivalent to $\pi$. The policy gradient update considered is defined below.

$$\pi_{k+1} := \mathbf{Proj}_\Pi \left[ \pi_k + \eta \frac{\partial \rho^\pi}{\partial \pi} \bigg|_{\pi = \pi_k} \right] \qquad \forall k \geq 0, \tag{6}$$

where $\mathbf{Proj}_\Pi$ denotes the orthogonal projection in the Euclidean norm onto the space of randomized policies $\Pi$ and $\eta$ denotes the step size of the update. In the following subsection, we recall the policy gradient result within the framework of discounted reward MDPs and address why it cannot be directly applied to the average reward scenario.

## 2.2 RELATIONSHIP TO DISCOUNTED REWARD MDPs

Let $\rho^\pi_{\mu,\gamma} := \mu^T (1 - \gamma \mathbb{P}^\pi)^{-1} r^\pi$ represent the discounted reward value function associated with policy $\pi$, under the initial distribution $\mu \in \Delta\mathcal{S}$ and where $\gamma$ represents the discount factor (Bertsekas, 2007). Consider the projected policy gradient update given below.

$$\pi_{k+1} := \mathbf{Proj}_\Pi \left[ \pi_k + \eta \frac{\partial \rho^\pi_{\mu,\gamma}}{\partial \pi} \bigg|_{\pi = \pi_k} \right] \qquad \forall k \geq 0. \tag{7}$$

When the step size $\eta = \frac{(1-\gamma)^3}{2\gamma|\mathcal{A}|}$, the iterates $\pi_k$ generated from projected gradient ascent equation 7 satisfy the following equation:

$$\rho^*_{\mu,\gamma} - \rho^{\pi_k}_{\mu,\gamma} \leq \frac{256|\mathcal{S}||\mathcal{A}|}{k(1-\gamma)^5} \left\| \frac{d^{\pi^*}_{\mu,\gamma}}{\mu} \right\|_\infty^2, \tag{8}$$

where $\rho^*_{\mu,\gamma}$ represents the optimal value function under initial distribution $\mu$, and $d^{\pi^*}_{\mu,\gamma} := (1 - \gamma)\mu^T(1 - \gamma\mathbb{P}^{\pi^*})^{-1}$ represents the state occupancy measure under optimal policy $\pi^*$ (Xiao, 2022b). Under some mild conditions the average reward $\rho^\pi$ associated with a policy $\pi$ and the value function $\rho^\pi_{\mu,\gamma}(s)$ are related as below:

$$\rho^\pi = \lim_{\gamma \to 1}(1 - \gamma)\rho^\pi_{\mu,\gamma}(s). \tag{9}$$

Note that the above relation (Bertsekas, 2007; Ross, 1983) holds for all $s \in \mathcal{S}$ and all $\mu \in \Delta\mathcal{S}$ since the average reward is independent of the initial state distribution. Upon leveraging the relation in equation 9 and multiplying equation 8 with $(1 - \gamma)$, it is apparent that the upper bound of equation 8 in the limit of $\gamma \to 1$ tends to infinity. This is due to $(1 - \gamma)^4$ that remains in the denominator of equation 8 upon multiplying with $(1 - \gamma)$. Therefore it is necessary to devise an alternative proof technique in order to analyze the global convergence of policy gradient in the context of average reward MDPs. Prior to presenting the main result and its proof, we state the assumption used in our analysis.

**Assumption 1.** *For every policy $\pi \in \Pi$, the transition matrix $\mathbb{P}^\pi$ associated with the induced Markov chain is irreducible and aperiodic. This assumption also means that there exist constants $C_e < \infty$ and $\lambda \in [0, 1)$ such that for any $k \in \mathbb{N}$ and any $\pi \in \Pi$, the Markov chain corresponding to $\mathbb{P}^\pi$ is geometrically ergodic i.e., $\| (\mathbb{P}^\pi)^k - \mathbb{1}(d^\pi)^\top \|_\infty \leq C_e\lambda^k$.*

## 3 MAIN RESULTS

**Theorem 1.** *Let $\rho^{\pi_k}$ be the average reward corresponding to the policy iterates $\pi_k$, obtained through the policy gradient update equation 6. Let $\rho^*$ represent the optimal average reward, that is, $\rho^* = \max_{\pi \in \Pi} \rho^\pi$. There exist constants $L_2^\Pi$ and $C_{PL}$, determined by the underlying MDP, such that when the step size $\eta < \frac{1}{L_2^\Pi}$, the following holds:*

* *For all MDPs it is true that,*

$$\rho^* - \rho^{\pi_k} \leq \frac{1}{\frac{1}{\rho^* - \rho^{\pi_0}} + \nu k}, \qquad \forall k \geq 0. \tag{10}$$

*where $\nu := \left( \frac{1}{32C_{PL}^2|\mathcal{S}|L_2^\Pi} \right)\left( 1 + 4\left( \frac{1}{32C_{PL}^2|\mathcal{S}|L_2^\Pi} \right) \right)^{-\frac{3}{2}}$*

* *For simple MDPs (i.e. $L_2^\Pi \ll 1$) we obtain exponential convergence, that is*

$$\rho^* - \rho^{\pi_k} \leq c^{-\frac{k}{2}}\left( \rho^* - \rho^{\pi_0} \right)^{\frac{1}{2^k}}, \qquad \forall k \geq 0 \tag{11}$$

*where $\frac{1}{c} = 32|\mathcal{S}|L_2^\Pi C_{PL}^2 < 1$.*

**Remark:** It is worth noting that the above bounds correspond to a regret of $O(\log(T))$. While regret is typically a concept associated with online learning settings, we present it here to facilitate the development of learning algorithms inspired by the findings of this work. All previous results on convergence for discounted MDPs are of the form $\frac{\sigma}{k^p}$ (Agarwal et al., 2020; Mei et al., 2022; Xiao, 2022a), where $\sigma$ is a large constant. However, since the worst sub-optimality is 1, the bound $\frac{\sigma}{k^p}$ becomes less meaningful for initial $k$. In contrast, our bound $\frac{1}{\frac{1}{\rho^* - \rho^{\pi_0}} + \nu k}$ is meaningful from the very first iteration. The maximum sub-optimality is $\rho^* - \rho^{\pi_0}$ at $k = 0$, and it decreases monotonically thereafter.

Our second result, an observation that is novel to this work, shows that simple MDPs exhibit much faster (linear) convergence rates. This explains why techniques like reward shaping, which simplify the MDP, can be highly effective.

### 3.1 KEY IDEAS AND PROOF OUTLINE

A similar result was proved for discounted reward MDPs in Agarwal et al. (2020); Xiao (2022a). An important property pivotal to the global convergence analysis of the projected policy gradient

is the smoothness of the discounted reward value function. Demonstrating the smoothness of the discounted reward value function is relatively straightforward due to the contractive properties of the discount factor. However, this poses a significant challenge in the context of average reward MDPs. Here, the absence of a discount factor as a source of contraction, coupled with the lack of uniqueness in the average reward value function, complicates the task of proving the smoothness of the average reward. Therefore, the first important property we prove is the smoothness of average reward.

### 3.1.1 SMOOTHNESS OF AVERAGE REWARD

A differentiable function $f : \mathcal{C} \to \mathbb{R}$ is called $L$-smooth if it satisfies

$$\|\nabla f(y) - \nabla f(x)\|_2 \leq L\|y - x\|_2 \qquad \forall y, x \in \mathcal{C}. \tag{12}$$

where $\mathcal{C}$ is some subset of $\mathbb{R}^n$. Further if the function is $L$-smooth, it satifies the following property.

$$\left| f(y) - f(x) - \langle \nabla f(x), y - x \rangle \right| \leq \frac{L}{2}\|y - x\|^2 \qquad \forall y, x \in \mathcal{C}, \tag{13}$$

From the above definition, it is apparent that if $f$ is $L$-smooth then $cf$ is $|c|L$-smooth for any $c \in \mathbb{R}$. It can be shown that the infinite horizon discounted reward $V^\pi_{\mu,\gamma}$ is $\frac{2\gamma|\mathcal{A}|}{(1-\gamma)^3}$-smooth. Leveraging the result in equation 9, one can see that the smoothness constant of $\rho^\pi$ is $\lim_{\gamma \to 1} \frac{2\gamma|\mathcal{A}|}{(1-\gamma)^2} \to \infty$. Hence, the smoothness of the discounted reward cannot be leveraged to show the smoothness of the average reward.

In this paper, we establish the smoothness of the infinite horizon average reward by first establishing the smoothness of the associated relative value function. We then leverage the average reward Bellman Equation 3 to establish the smoothness of the average reward in terms of the smoothness of the relative value function. However, since the relative value function is unique up to an additive constant, we consider the projection of the value function onto the subspace orthogonal to the $\mathbb{1}$ vector. This provides us with an unique representation of the value function whose smoothness can be evaluated.

Let $\Phi \in \mathbb{R}^{|\mathcal{S}| \times |\mathcal{S}|}$ be the projection matrix that maps any vector to its orthogonal projection in the Euclidean norm onto the subspace perpendicular to the $\mathbb{1}$ vector. Then the following lemma holds.

**Lemma 1.** *Let $I$ be the identity matrix and $\mathbb{1}$ be the all ones vector, both of dimension $|\mathcal{S}|$. Then the orthogonal projection matrix is given by $\Phi = \left( I - \frac{\mathbb{1}\mathbb{1}^\top}{|\mathcal{S}|} \right)$. The unique value function $v^\pi_\phi$ is obtained as a solution to the following fixed point equation,*

$$v^\pi_\phi = \Phi \left( r^\pi + \mathbb{P}^\pi v^\pi_\phi - \rho^\pi \mathbb{1} \right) \tag{14}$$

*and can be alternatively represented as*

$$v^\pi_\phi = \left( I - \Phi\mathbb{P}^\pi \right)^{-1} \Phi r^\pi. \tag{15}$$

Since $\mathbb{P}^\pi$ has 1 as its Perron Frobenius eigenvalue, $(I - \mathbb{P}^\pi)$ is a singular matrix. It can be verified that $\Phi\mathbb{1} = 0$, hence $\mathbb{1}$ is an eigenvector of $\Phi\mathbb{P}^\pi$ for all $\pi$ with a corresponding eigenvalue of 0. It can subsequently be proven that the rest of the eigenvalues of $\Phi\mathbb{P}^\pi$ are all less than one in terms of their absolute value and hence equation 15 is well defined.

With a unique closed form for the average reward value function established, the subsequent task is to determine its smoothness constant. Given that the smoothness constant of a function $f$ corresponds to the largest eigenvalue of its Hessian, we adopt an analytical approach similar to that presented in Agarwal et al. (2020). This involves utilizing directional derivatives and evaluating the maximum rate of change of derivatives across all directions within the policy space. It's important to note that since we are maximizing over directions expressible as differences between any two policies within the policy space, the resulting Lipschitz and smoothness constants are referred to as the restricted Lipschitz and smoothness constants, respectively. The restricted smoothness of the average reward value function is stated below.

**Lemma 2.** *For any policy $\pi \in \Pi$, there exist constants $C_m, C_p, C_r, \kappa_r \in \mathbb{R}^+$ which are determined by the underlying MDP, such that the value function $v^\pi_\phi$ is $4\left( 2C_m^3 C_p^2 \kappa_r + C_m^2 C_p C_r \right)$-smooth.*

Since the average reward value function is Lipschitz and smooth with respect to its policy, one can directly utilize this property to establish the Lipschitzness and smoothness of the average reward. These results are characterized in the following lemmas.

**Lemma 3.** *For any policy $\pi \in \Pi$, there exist constants $C_m, C_p, C_r, \kappa_r \in \mathbb{R}^+$ which are determined by the underlying MDP, such that the average reward $\rho^\pi$ is $L_1^\Pi$-Lipschitz.*

$$\left| \left\langle \frac{\partial \rho^\pi}{\partial \pi}, \pi' - \pi \right\rangle \right| \leq L_1^\Pi \|\pi' - \pi\|_2, \qquad \forall \pi, \pi' \in \Pi, \tag{16}$$

*where $L_1^\Pi = 2(C_r + C_p C_m \kappa_r + 2(C_m^2 C_p \kappa_r + C_m C_r))$*

The restricted Lipschitzness of the average reward is utilized to prove its restricted smoothness.

**Lemma 4.** *For any policy $\pi \in \Pi$, there exist constants $C_m, C_p, C_r, \kappa_r \in \mathbb{R}^+$ which are determined by the underlying MDP, such that the average reward $\rho^\pi$ is $L_2^\Pi$-smooth.*

$$\left| \left\langle \pi' - \pi, \frac{\partial^2 \rho^\pi}{\partial \pi^2}(\pi' - \pi) \right\rangle \right| \leq \frac{L_2^\Pi}{2} \|\pi' - \pi\|_2^2 \qquad \forall \pi, \pi' \in \Pi, \tag{17}$$

*where $L_2^\Pi = 4(C_p^2 C_m^2 \kappa_r + C_p C_m C_r + (C_p + 1)(C_m^2 C_p \kappa_r + C_m C_r) + 4(C_m^3 C_p^2 \kappa_r + C_m^2 C_p C_r))$.*

Note that the restricted Lipschitz constant of the average reward is upper bounded by its general Lipschitz constant:

$$\max_{\pi' \in \Pi : \|\pi' - \pi\|_2 \leq 1} \left| \left\langle \frac{\partial \rho^\pi}{\partial \pi}, \pi' - \pi \right\rangle \right| \leq \max_{u \in \mathbb{R}^{S \times A} : \|u\|_2 \leq 1} \left| \left\langle \frac{\partial \rho^\pi}{\partial \pi}, u \right\rangle \right| \tag{18}$$

By confining our analysis of the smoothness constants to the policy class, we introduce a dependency of our convergence bounds on MDP-specific constants, including $C_r, C_p, C_m, C_e$ and $\kappa_r$. These constants capture the complexity of the underlying MDP and are exclusive to the analysis presented in this paper, as there appears to be no such dependency observed in the global convergence bounds of Agarwal et al. (2020). A more detailed description of these constants can be found in Table 1, where $k_1$ and $k_2$ represent MDP-independent numeric constants. These constants, which rely on the characteristics of the MDP, suggest that the projected policy gradient may achieve faster convergence in MDPs with lower complexity as opposed to those with higher complexity. The range of these constants can be found in Appendix A. We now proceed to analyze the convergence of projected policy gradient utilizing the smoothness of the average reward.

### 3.1.2 CONVERGENCE OF POLICY GRADIENT

Using the smoothness property of the average reward, it is possible to show that the improvement in the successive average reward iterates is bounded from below by the product of the smoothness constant and the difference in the policy iterates, as described in the lemma below.

**Lemma 5.** *Let $\rho^{\pi_k}$ be the average reward corresponding to the policy iterate $\pi_k$ obtained from equation 6. Let $L_2^\Pi$ be as in Lemma 4. Then,*

$$\rho^{\pi_{k+1}} - \rho^{\pi_k} \geq \frac{L_2^\Pi}{2} \|\pi_{k+1} - \pi_k\|^2, \qquad \forall k \in \mathbb{N}. \tag{19}$$

Successively increasing iterates are not sufficient to guarantee finite time global convergence bounds. It is therefore necessary to bound the suboptimality associated with each iterate. We do so by leveraging the performance difference lemma stated below.

**Lemma 6.** *Let $\rho^*$ be the globally optimal average reward. Then for any $\pi \in \Pi$, the suboptimality of $\rho^\pi$ can be expressed as:*

$$\rho^* - \rho^\pi = \sum_s d^{\pi^*}(s) \sum_a Q^\pi(s, a)[\pi^*(a|s) - \pi(a|s)]. \tag{20}$$

*Proof.* The proof can be found in Cao (1999). $\qquad\square$

In the next lemma, we upper bound the right-hand side of equation 20 in terms of the gradient of $\rho^\pi$.

Table 1: Constants capturing the MDP Complexity

| | Definition | Range | Remark |
|---|---|---|---|
| $C_m$ | $\max_{\pi\in\Pi}\|(I-\Phi\mathbb{P}^\pi)^{-1}\|$ | $\frac{2C_e\|\mathcal{S}\|}{1-\lambda}$ See Assumption 1 for definition of $C_e$ and $\lambda$ | Lowest rate of mixing |
| $C_p$ | $\max_{\pi,\pi'\in\Pi}\frac{\|\mathbb{P}^{\pi'}-\mathbb{P}^\pi\|}{\|\pi'-\pi\|_2}$ | $[0,\sqrt{\|\mathcal{A}\|}]$ | Diameter of transition kernel |
| $C_r$ | $\max_{\pi,\pi'}\frac{\|r^{\pi'}-r^\pi\|_\infty}{\|\pi'-\pi\|_2}$ | $[0,\sqrt{\|\mathcal{A}\|}]$ | Diameter of reward function |
| $\kappa_r$ | $\max_\pi\|\Phi r^\pi\|_\infty$ | $[0,2)$ | Variance of reward function |
| $\frac{L_1^\Pi}{2}$ | $C_r + C_pC_m\kappa_r + 2(C_m^2C_p\kappa_r+C_mC_r)$ | $[0,k_1\sqrt{\|\mathcal{A}\|}C_m^2]$ | Restricted Lipschitz constant |
| $\frac{L_2^\Pi}{4}$ | $C_p^2C_m^2\kappa_r+C_pC_mC_r+(C_p+1)(C_m^2C_p\kappa_r+C_mC_r)+4(C_m^3C_p^2\kappa_r+C_m^2C_pC_r)$ | $[0,k_2\|\mathcal{A}\|C_m^3]$ | Restricted smoothness constant |

$C_p, C_m$ are defined using operator norm w.r.t. $L_\infty$ norm. Precisely,
$$C_m = \max_\pi \max_{\|v\|_\infty\leq 1}\|(I-\Phi P^\pi)^{-1}v\|_\infty, \text{ and } C_p = \max_{\pi,\pi'\in\Pi}\max_{\|v\|_\infty\leq 1}\frac{\|(P^{\pi'}-P^\pi)v\|_\infty}{\|\pi'-\pi\|_2}$$

**Lemma 7.** *The suboptimality of any $\pi\in\Pi$ satisfies:*

$$\rho^* - \rho^\pi \leq C_{PL}\max_{\pi'\in\Pi}\left\langle\pi'-\pi,\frac{\partial\rho^\pi}{\partial\pi}\right\rangle, \qquad \forall\pi\in\Pi, \tag{21}$$

*where $C_{PL} = \max_{\substack{\pi\in\Pi\\s\in\mathcal{S}}}\frac{d^{\pi^*}(s)}{d^\pi(s)}$.*

Note that $C_{PL}$ is a constant that is proportional to the size of the state space. We do not know if the appearance of such a constant is inevitable or not; however, it should be noted such a constant appears in prior works on discounted reward problems as well (Agarwal et al., 2020; Xiao, 2022a).

It is possible to further upper bound the expression in Lemma 7 using the smoothness property of the average reward.

**Lemma 8.** *Let $\pi_k$ be the policy iterates generated by equation 6. Then for all $\pi'\in\Pi$ it is true that,*

$$\left\langle\frac{\partial\rho_{\pi_{k+1}}}{\partial\pi_{k+1}},\pi'-\pi_{k+1}\right\rangle \leq 4\sqrt{\|\mathcal{S}\|}L_2^\Pi\|\pi_{k+1}-\pi_k\|, \tag{22}$$

Lemmas 5,7 and 8 are combined to prove the result in Theorem 1.

### 3.2 EXTENSION TO DISCOUNTED REWARD MDPS

Existing performance bounds in the context of discounted reward MDPs require an iteration complexity of $O\left(\frac{\|\mathcal{S}\|\|\mathcal{A}\|}{(1-\gamma)^5\epsilon}\right)$ to achieve policies with suboptimality of $\epsilon$ (Xiao, 2022a). These bounds are independent of the hardness of the underlying MDP. Our approach improves on this bound, yielding an $O(\frac{\|\mathcal{S}\|L_2^\Pi}{\epsilon})$ iteration complexity, where $L_2^\Pi = C_p^2\widehat{C}_m^2\kappa_r + C_p\widehat{C}_mC_r + (C_p+1)(\widehat{C}_m^2C_p\kappa_r + \widehat{C}_mC_r) + 4(\widehat{C}_m^3C_p^2\kappa_r + \widehat{C}_m^2C_pC_r)$, where $\widehat{C}_m := \|(I-\gamma\mathbb{P}^\pi)^{-1}\|$. It is straightforward to see that $\widehat{C}_m \leq \frac{1}{1-\gamma}$. Hence, the iteration complexity improves to $O\left(\frac{L_2^\Pi\|\mathcal{S}\|}{(1-\gamma)^5\epsilon}\right)$, as the constants satisfy $\kappa_r \leq 2$ and $C_p, C_r \leq \sqrt{\|\mathcal{A}\|}$. Further, the approach considered in this paper provides faster convergence rates for MDPs with low complexity, i.e., MDPs that have low values of $C_p$ or $C_r$. The exact

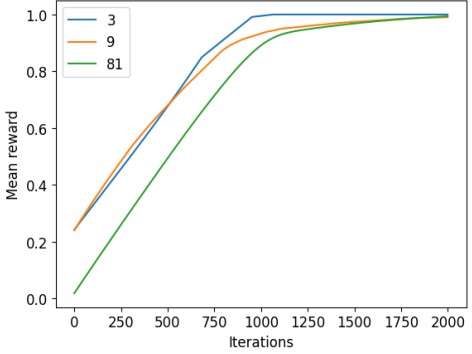 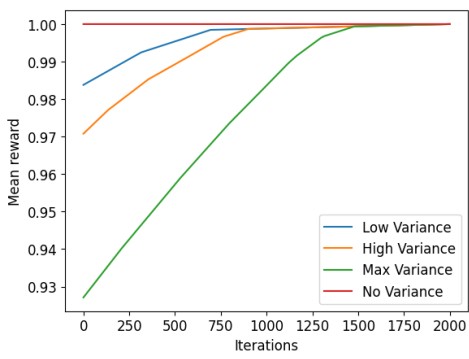

(a) Convergence as a function of state space cardinality

(b) Convergence as a function of reward variance

Figure 1: Improvement in average reward as a function of MDP complexity

performance bounds can be obtained from an approach similar to the one outlined in Kumar et al. (2023), where $L_2^\Pi$ represents the restricted smoothness constant of the discounted return $\rho_\gamma^\pi$. This constant can be derived through a process analogous to the one described in this paper.

For instance, consider a trivial MDP for which $C_p = 0$ or $\kappa_r = 0$ (implies $C_r = 0$), i.e., an MDP where the transition kernel is independent of the action enacted. For this trivial MDP every policy is an optimal policy. The state of the art convergence guarantees (Xiao, 2022a), still requires $O(|\mathcal{S}||\mathcal{A}|\epsilon^{-1})$ iterations for $\epsilon$ close optimal policy. Whereas, the performance bounds presented in this paper predict $O(|\mathcal{S}|\epsilon^{-1})$ iterations for convergence. Thus the constant $L_2^\Pi$ captures the hardness of the MDP. Therefore, MDPs with lower complexity, i.e., lower values of $L_2^\Pi$, converge faster than MDPs with higher complexity, thus improving on current complexity-independent bounds.

## 4 SIMULATIONS

Here, we present simulations corresponding to two MDP complexity measures. In order to study the convergence of projected policy gradient in the context of average reward MDPs and its dependence on the underlying MDP complexity, we present simulation results corresponding to two complexity measures: the cardinality of state and action spaces, and the diameter of the reward function.

Figure 1(a) considers MDPs with $(|\mathcal{S}|, |\mathcal{A}|) = \{(3, 3), (9, 9), (81, 81)\}$. We construct the transition kernel and the reward function in the same manner for all MDPs, which we discuss in the appendix. Projected policy gradient was implemented for 2000 iterations and the overall average reward is plotted as a function of iteration number. As expected, the convergence rate is slower when $(|\mathcal{S}|, |\mathcal{A}|)$ are larger due to the fact that the reward smoothness constant is larger. This reduction stems from small values of $C_M, C_r, C_p$, which are characteristic of MDPs with smaller state and action space cardinalities when the transition kernel and reward structures are similar. A less obvious result is that, even for MDPs with a fixed cardinality of state and action spaces, the rate of convergence can be considerably different as shown in Figure 1(b). For this simulation, we fix the state and action space cardinality at $(|\mathcal{S}|, |\mathcal{A}|) = (16, 16)$. We randomly generate a transition kernel, which remains constant across different single-step reward functions corresponding to varying reward variances. In particular, we consider four different reward variances - no variance, low, high and maximal variance. We recall the definition of $C_r$ found in Table 1. We see that $C_r$ scales with reward variance. Specifically, $C_r$ is large when small changes to the policy result in significant modifications to the mean reward. Therefore, we anticipate that higher reward variance will lead to slower convergence. Additional details are found in the appendix. The observed convergence trend aligns with the theoretical bounds obtained, indicating that MDPs with small values of $C_r$ tend to converge relatively faster.

Next, we discuss the impact of $C_p$ on convergence of policy gradient. We consider MDPs of size 16, i.e., $(|\mathcal{S}|, |\mathcal{A}|) = (16, 16)$. We generate three different transition kernels. The first is uniform, so the actions do not change the transition probabilities, the second is deterministic (i.e., there exists

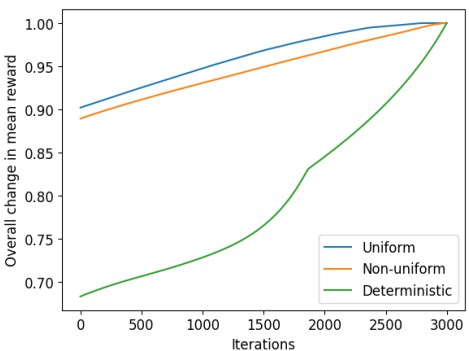

Figure 2: Convergence as a function of $C_p$

some $s' \in \mathcal{S}$ such that $\mathbb{P}(s'|s,a) = 1$ for all $(s,a) \in (\mathcal{S}, \mathcal{A})$), and the last is non-uniform but stochastic transition kernel. We recall the definition of $C_p$ from Table 1. We see that $C_p$ is larger when the transition probabilities change by a greater amount with small changes to the policy. Thus, deterministic MDPs should have higher $C_p$ values then ones that are more stochastic. Additional details are provided in the appendix. We run the policy gradient algorithm considered in this paper for each MDP setting and plot the overall change in average reward as a function of iterations, for 3000 iterations. Figure 2 indicates that policy gradient in MDPs corresponding to small values of $C_p$ converges relatively faster than for MDPs corresponding to large values of $C_p$. Hence, the performance bounds obtained in Theorem 1 are in some sense, more representative of the empirical convergence trend of the policy gradient algorithm.

**Note on Limitations and Future Work:** Although we study tabular policies, this approach can be generalized to parametric class of policies. This study uses the exact value of the gradient and does not account for learning errors in the analysis. Nonetheless, we highlight that this is the first comprehensive proof of global convergence for policy gradient methods in average reward MDPs. Future work will focus on incorporating learning errors into this framework.

## 5 CONCLUSION

In this paper, we presented the first comprehensive finite-time global convergence analysis of policy gradient for infinite horizon average reward MDPs. Key contributions include eliminating the smoothness assumption from previous work, deriving an explicit expression for the smooth average cost, and proving sublinear convergence with a regret of $O(\log(T))$. Our findings offer a more general and robust understanding of policy gradient methods in average reward MDPs, addressing long-standing challenges such as the lack of uniqueness in value functions. We also extended our analysis to discounted reward MDPs, providing stronger performance bounds by incorporating a complexity parameter beyond state and action space sizes. The theoretical results were further supported by simulations that highlighted how the structure of the underlying MDP influences convergence rates. These insights open avenues for refining performance bounds and exploring real-world applications in both average and discounted reward MDPs.

## ACKNOWLEDGEMENTS

Research conducted by Y.M. and R.S. was supported in part by NSF Grants CNS 23-12714, CCF 22-07547, CNS 21-06801, and AFOSR Grant FA9550-24-1-0002.

This research was also supported by the Israel Science Foundation (Grants No. 2199/20 and 3109/24).

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

## A SMOOTHNESS OF AVERAGE REWARD

### A.1 PROOF OF LEMMA 1

Consider the subspace orthogonal $E$ to the all ones vector $\mathbb{1} \in \mathbb{R}^{|\mathcal{S}|}$ defined below:

$$E = \text{span} \left\{ \theta \in \mathbb{R}^{|\mathcal{S}|} : \theta^\top \mathbb{1} = 0. \right\} \tag{23}$$

The orthogonal projection $v_\phi$ of a vector $v$ in the Euclidean norm onto the subspace $E$ is defined as:

$$v_\phi = \underset{u \in \mathbb{E}}{\arg \min} \, ||v - u||_2 \tag{24}$$

It can be checked that the closed form expression for $v_\phi$ is given by:

$$v_\phi = \left( I - \frac{\mathbb{1}\mathbb{1}^\top}{|\mathcal{S}|} \right) v \tag{25}$$

where $I \in \mathbb{R}^{|\mathcal{S}| \times |\mathcal{S}|}$ is the identity matrix.

Consider the projection of the vector $r^\pi + \mathbb{P}^\pi v^\pi - \rho^\pi \mathbb{1}$ onto $E$ for any policy $\pi \in \Pi$. The above projection is identical to the projection of $r^\pi + \mathbb{P}^\pi v^\pi$ onto $E$, since $\rho^\pi \mathbb{1}$ lies in the nullspace of $\Phi$.

$$\Phi(r^\pi - \rho^\pi \mathbb{1} + \mathbb{P}^\pi v) = (r^\pi - \rho^\pi \mathbb{1} + \mathbb{P}^\pi v) - \left\langle \mathbb{1}, r^\pi - \rho^\pi \mathbb{1} + \mathbb{P}^\pi v \right\rangle \frac{\mathbb{1}}{|\mathcal{S}|} \tag{26}$$

$$= (r^\pi + \mathbb{P}^\pi v) - \left\langle \mathbb{1}, r^\pi + \mathbb{P}^\pi v \right\rangle \frac{\mathbb{1}}{|\mathcal{S}|} \tag{27}$$

$$= r^\pi - \langle r^\pi, \mathbb{1} \rangle \frac{\mathbb{1}}{|\mathcal{S}|} + \mathbb{P}^\pi v - \langle \mathbb{1}, \mathbb{P}^\pi v \rangle \frac{\mathbb{1}}{|\mathcal{S}|} \tag{28}$$

$$= r^\pi - \langle r^\pi, \mathbb{1} \rangle \frac{\mathbb{1}}{|\mathcal{S}|} + \mathbb{P}^\pi v - \frac{\mathbb{1}}{|\mathcal{S}|} (\mathbb{1}^\top \mathbb{P}^\pi v) \tag{29}$$

$$= (I - \frac{\mathbb{1}\mathbb{1}^\top}{|\mathcal{S}|}) r^\pi + (I - \frac{\mathbb{1}\mathbb{1}^\top}{|\mathcal{S}|}) \mathbb{P}^\pi v \tag{30}$$

$$= \Phi \left[ r^\pi + \mathbb{P}^\pi v \right]. \tag{31}$$

Consider the average reward Bellman equation corresponding to policy $\pi \in \Pi$:

$$\rho^\pi \mathbb{1} + v^\pi = r^\pi + \mathbb{P}^\pi v^\pi \tag{32}$$

Imposing an additional constraint ${v^\pi}^\top \mathbb{1} = 0$ yields a unique average reward value function denoted by $v_\phi^\pi$. Moreover, it is true that,

$$\Phi v_\phi^\pi + \Phi \rho^\pi \mathbb{1} = \Phi r^\pi + \Phi \mathbb{P}^\pi v_\phi^\pi \tag{33}$$

$$\implies \Phi v_\phi^\pi = \Phi r^\pi + \Phi \mathbb{P}^\pi v_\phi^\pi, \tag{34}$$

$$\overset{(a)}{\implies} v_\phi^\pi = \Phi [r^\pi + \mathbb{P}^\pi v_\phi^\pi], \tag{35}$$

where (a) is true because ${v^\pi}^\top \mathbb{1} = 0 \implies \Phi v^\pi = v^\pi$. Thus the projected value function with an unique representation is given by:

$$v_\phi^\pi = [I - \Phi \mathbb{P}^\pi]^{-1} \Phi r^\pi, \tag{36}$$

and the existence of the inverse is proven in Subsection A.2, Lemma 12. An alternate expression for the projected value function is given by: $v_\phi^\pi = \left( I + \mathbb{1}\mathbb{1}^\top D - \frac{\mathbb{1}\mathbb{1}^\top}{|\mathcal{S}|} \right) v_0^\pi$, where $D \in \mathbb{R}^{|\mathcal{S}| \times |\mathcal{S}|}$ is a diagonal matrix whose entries correspond to the stationary measure over the states associated with policy $\pi$. See Tsitsiklis & Van Roy (1999) for more details.

### A.2 PROOF THAT EIGENVALUES OF $(I - \Phi \mathbb{P}^\pi)$ ARE NON-ZERO

In this subsection, we introduce the lemmas required to establish the proof of the eigenvalues of $\left( I - \frac{\mathbb{1}\mathbb{1}^\top}{|\mathcal{S}|} \right) \mathbb{P}^\pi$ being nonzero. We use the following notation: $\mathbb{1} \in \mathbb{R}^n$ represents the all ones vector and $I \in \mathbb{R}^{n \times n}$ is the identity matrix.

**Lemma 9.** *Let $A \in \mathbb{R}^{n \times n}$ be a stochastic matrix. It is true that*

$$\left( \left( I - \frac{\mathbb{1}\mathbb{1}^\top}{n} \right) A \right)^k = \left( I - \frac{\mathbb{1}\mathbb{1}^\top}{n} \right) A^k. \tag{37}$$

*Proof.* For any $k \in \mathbb{N}$, consider,

$$\left( I - \frac{\mathbb{1}\mathbb{1}^\top}{n} \right) A^k \left( I - \frac{\mathbb{1}\mathbb{1}^\top}{n} \right) A = \left( A^k - \frac{\mathbb{1}\mathbb{1}^\top}{n} A^k \right) \left( A - \frac{\mathbb{1}\mathbb{1}^\top}{n} A \right) \tag{38}$$

$$= A^{k+1} - \frac{\mathbb{1}\mathbb{1}^\top}{n} A^{k+1} - A^k \frac{\mathbb{1}\mathbb{1}^\top}{n} A + \frac{\mathbb{1}\mathbb{1}^\top}{n} A^k \frac{\mathbb{1}\mathbb{1}^\top}{n} A \tag{39}$$

$$\overset{(a)}{=} A^{k+1} - \frac{\mathbb{1}\mathbb{1}^\top}{n} A^{k+1} - \frac{\mathbb{1}\mathbb{1}^\top}{n} A + \frac{\mathbb{1}\mathbb{1}^\top}{n} \frac{\mathbb{1}\mathbb{1}^\top}{n} A, \tag{40}$$

$$\overset{(b)}{=} A^{k+1} - \frac{\mathbb{1}\mathbb{1}^\top}{n} A^{k+1} - \frac{\mathbb{1}\mathbb{1}^\top}{n} A + \frac{\mathbb{1}\mathbb{1}^\top}{n} A, \tag{41}$$

$$= A^{k+1} - \frac{\mathbb{1}\mathbb{1}^\top}{n} A^{k+1}. \tag{42}$$

$$= \left( I - \frac{\mathbb{1}\mathbb{1}^\top}{n} \right) A^{k+1} \tag{43}$$

where (a) is true because $A^k \mathbb{1} = \mathbb{1}$ and (b) follows from the fact that $\frac{\mathbb{1}^\top \mathbb{1}}{n} = \mathbb{1}$. From mathematical induction it thus follows that,

$$\left( \left( I - \frac{\mathbb{1}\mathbb{1}^\top}{n} \right) A \right)^k = \left( I - \frac{\mathbb{1}\mathbb{1}^\top}{n} \right) A^k \qquad \forall k \in \mathbb{N}. \tag{44}$$

$\square$

**Lemma 10.** *For any irreducible and aperiodic stochastic matrix $A \in \mathbb{R}^{n \times n}$, it is true that*

$$\lim_{k \to \infty} \left( \left( I - \frac{\mathbb{1}\mathbb{1}^\top}{n} \right) A \right)^k = 0 \tag{45}$$

*Proof.* From Lemma 9 we have,

$$\left( \left( I - \frac{\mathbb{1}\mathbb{1}^\top}{n} \right) A \right)^k = \left( I - \frac{\mathbb{1}\mathbb{1}^\top}{n} \right) A^k \qquad \forall k \in \mathbb{N} \tag{46}$$

Since $A$ is irreducible and aperiodic, the following limit converges to the stationary distribution $d \in \mathbb{R}_+^n$ associated with $A$.

$$\lim_{k \to \infty} A^k = \mathbb{1} d^\top \tag{47}$$

Consider the following,

$$\lim_{k \to \infty} \left( \left( I - \frac{\mathbb{1}\mathbb{1}^\top}{n} \right) A \right)^k = \lim_{k \to \infty} \left( I - \frac{\mathbb{1}\mathbb{1}^\top}{n} \right) A^k \tag{48}$$

$$= \left( I - \frac{\mathbb{1}\mathbb{1}^\top}{n} \right) \lim_{k \to \infty} A^k \tag{49}$$

$$= \left( I - \frac{\mathbb{1}\mathbb{1}^\top}{n} \right) \mathbb{1} d^\top \qquad \text{(from Equation equation 47),} \tag{50}$$

$$= \mathbb{1} d^\top - \frac{\mathbb{1}\mathbb{1}^\top}{n} \mathbb{1} d^\top \tag{51}$$

$$\overset{(a)}{=} \mathbb{1} d^\top - \mathbb{1} d^\top \tag{52}$$

$$= 0. \tag{53}$$

where (a) is true because $\frac{\mathbb{1}^\top \mathbb{1}}{n} = 1$.

$\square$

**Lemma 11.** *Let $A \in \mathbb{R}^{n \times n}$ be a matrix such that $\lim_{k \to \infty} A^k = 0$. Then $(I - A)^{-1} = \sum_{k=0}^{\infty} A^k$.*

*Proof.* For any $K \in \mathbb{N}$, consider the following,

$$(I - A) \left( \sum_{k=0}^{K} A^k \right) = I - A^{K+1}, \tag{54}$$

$$\implies (I - A) \left( \lim_{K \to \infty} \sum_{k=0}^{K} A^k \right) = \lim_{K \to \infty} \left( I - A^{K+1} \right) \overset{(a)}{=} I, \tag{55}$$

where (a) follows from the fact that $\lim_{k \to \infty} A^k = 0$. Hence the inverse of $(I - A)$ can be expressed as $(I - A)^{-1} = \sum_{k=0}^{\infty} A^k$. $\qquad\square$

**Lemma 12.** *Let $A \in \mathbb{R}^{n \times n}$ be an irreducible and aperiodic stochastic matrix. Then the matrix $\left( I - \left( I - \frac{\mathbb{1}\mathbb{1}^\top}{n} \right) A \right)$ is invertible and its inverse is given by:*

$$\left( I - \left( I - \frac{\mathbb{1}\mathbb{1}^\top}{n} \right) A \right)^{-1} = \sum_{k=0}^{\infty} \left( I - \frac{\mathbb{1}\mathbb{1}^\top}{n} \right) A^k \tag{56}$$

*Proof.* Let $\lambda_i$ be eigenvalues of $\left( I - \frac{\mathbb{1}\mathbb{1}^\top}{n} \right) A$. Then $\lambda_i^k$ represents the eigenvalues of $\left( \left( I - \frac{\mathbb{1}\mathbb{1}^\top}{n} \right) A \right)^k$. But from Lemma 10, we know that

$$\lim_{k \to \infty} \left( \left( I - \frac{\mathbb{1}\mathbb{1}^\top}{n} \right) A \right)^k = 0 \tag{57}$$

Since eigenvalues are continuous functions of their corresponding matrices and all eigenvalues of a zero matrix are zero, we thus have,

$$\lim_{k \to \infty} \lambda_i^k = 0 \qquad \forall i \in \{1, \dots, n\} \tag{58}$$

Equation 58 thus implies that $|\lambda_i| < 1, \forall i \in \{1, \dots, n\}$. Hence the matrix $\left( I - \left( \left( I - \frac{\mathbb{1}\mathbb{1}^\top}{n} \right) A \right) \right)$ has all non zero eigenvalues and is thus invertible. From Lemma 11, we know that

$$(I - A)^{-1} = \sum_{k=0}^{\infty} A^k \tag{59}$$

when $\lim_{k \to \infty} A^k = 0$. Since, $\lim_{k \to \infty} \left( \left( I - \frac{\mathbb{1}\mathbb{1}^\top}{n} \right) A \right)^k = 0$ from Lemma 10, we have the following result,

$$\left( I - \left( I - \frac{\mathbb{1}\mathbb{1}^\top}{n} \right) A \right)^{-1} = \sum_{k=0}^{\infty} \left( I - \frac{\mathbb{1}\mathbb{1}^\top}{n} \right) A^k \tag{60}$$

$\qquad\square$

From definition we have $\Phi = \left( I - \frac{\mathbb{1}\mathbb{1}^\top}{n} \right)$. Hence the inverse $(1 - \Phi \mathbb{P}^\pi)^{-1} = \sum_{k=0}^{\infty} \Phi \left( \mathbb{P}^\pi \right)^k$ exists and is well defined for all $\pi \in \Pi$.

## A.3 SMOOTHNESS OF THE AVERAGE REWARD VALUE FUNCTION $v_\phi^\pi$

In order to prove the smoothness of the average reward value function and the infinite horizon average reward, we consider an analysis inspired by Agarwal et al. (2020), where instead of computing the maximum eigenvalue of the associated Hessian matrices, we consider the maximum value of the directional derivative across all directions within the policy class.

Let $\pi, \pi' \in \Pi$ be any policies within the policy class. Then define $\pi_\alpha$ as a convex combination of policies $\pi$ and $\pi'$. That is

$$\pi_\alpha := (1-\alpha)\pi + \alpha\pi' \tag{61}$$

$$= \pi + \alpha(\pi' - \pi) \tag{62}$$

$$= \pi + \alpha u \tag{63}$$

where $u = \pi' - \pi$.

Since $\pi_\alpha$ is linear in $\alpha$, it is true that

$$\nabla_\alpha \pi_\alpha = \frac{d(\pi + \alpha u)}{d\alpha} = u, \qquad \text{and} \qquad \nabla_\alpha^2 \pi_\alpha = 0. \tag{64}$$

This thus implies,

$$\|\nabla_\alpha \pi_\alpha\|_2 = \|u\|_2 \leq \|\pi' - \pi\|_1 \leq 2S, \qquad \text{and} \qquad \|\nabla_\alpha^2 \pi_\alpha\|_2 = 0, \tag{65}$$

Thus, $\pi_\alpha$ is both $\|u\|_2$-Lipschitz and 0-smooth with respect to $\alpha$, for all $u$ that can be represented as the difference of any two policies.

From the definition of $\mathbb{P}^\pi$, we have

$$\mathbb{P}^{\pi_\alpha}(s'|s) = \sum_{a \in \mathcal{A}} \pi_\alpha(a|s)\mathbb{P}(s'|s,a) \tag{66}$$

$$= \sum_{a \in \mathcal{A}} [\pi(a|s) + \alpha u(a|s)] \, \mathbb{P}(s'|s,a) \tag{67}$$

$$\implies \frac{\partial \mathbb{P}^{\pi_\alpha}(s'|s)}{\partial \alpha} = \sum_{a \in \mathcal{A}} u(a|s)\mathbb{P}(s'|s,a). \tag{68}$$

That is,

$$\nabla_\alpha \mathbb{P}^{\pi_\alpha} = \mathbb{P}^u, \qquad \text{consequently} \qquad \nabla_\alpha^2 \mathbb{P}^{\pi_\alpha} = 0. \tag{69}$$

From the definition of $r^\pi$, we have

$$r^{\pi_\alpha}(s) = \sum_{a \in \mathcal{A}} \pi_\alpha(a|s)r(s,a) \tag{70}$$

$$= \sum_{a \in \mathcal{A}} [\pi(a|s) + \alpha u(a|s)] \, r(s,a) \tag{71}$$

$$\implies \frac{\partial r^{\pi_\alpha}(s)}{\partial \alpha} = \sum_{a \in \mathcal{A}} u(a|s)r(s,a). \tag{72}$$

That is,

$$\nabla_\alpha r^{\pi_\alpha} = r^u, \qquad \text{consequently} \qquad \nabla_\alpha^2 r^{\pi_\alpha} = 0. \tag{73}$$

Hence the policy $\pi_\alpha$, the associated reward $r^{\pi_\alpha}$ and the transition kernel $\mathbb{P}^{\pi_\alpha}$ are all Lipschitz and smooth with respect to $\alpha$.

**Lemma 13.** *Let $A(\alpha) \in \mathbb{R}^{n \times n}$ be a matrix such that $(I - A(\alpha))$ is invertible for all $\alpha \in [0,1]$. Define $M(\alpha) := (I - A(\alpha))^{-1}$. Then it is true that,*

$$\frac{\partial^2 M(\alpha)}{\partial \alpha^2} = \frac{\partial M(\alpha)}{\partial \alpha}\frac{\partial A(\alpha)}{\partial \alpha}M(\alpha) + M(\alpha)\frac{\partial^2 A(\alpha)}{\partial \alpha^2}M(\alpha) + M(\alpha)\frac{\partial A(\alpha)}{\partial \alpha}\frac{\partial M(\alpha)}{\partial \alpha}. \tag{74}$$

*Proof.*

$$M(\alpha)\,(I - A(\alpha)) = I \tag{75}$$

$$\frac{\partial M(\alpha)}{\partial \alpha}\,(I - A(\alpha)) - M(\alpha)\frac{\partial A(\alpha)}{\partial \alpha} = 0 \tag{76}$$

$$\frac{\partial M(\alpha)}{\partial \alpha} = M(\alpha)\frac{\partial A(\alpha)}{\partial \alpha}M(\alpha) \tag{77}$$

$$\frac{\partial^2 M(\alpha)}{\partial \alpha^2} = \frac{\partial}{\partial \alpha}\left(M(\alpha)\frac{\partial A(\alpha)}{\partial \alpha}M(\alpha)\right), \tag{78}$$

$$= \frac{\partial M(\alpha)}{\partial \alpha}\frac{\partial A(\alpha)}{\partial \alpha}M(\alpha) + M(\alpha)\frac{\partial^2 A(\alpha)}{\partial \alpha^2}M(\alpha) + M(\alpha)\frac{\partial A(\alpha)}{\partial \alpha}\frac{\partial M(\alpha)}{\partial \alpha} \tag{79}$$

$$\square$$

Consider the following definition utilized in the proofs of the upcoming lemmas.

$$M^{\pi_\alpha} = [I - \Phi\mathbb{P}^{\pi_\alpha}]^{-1} \tag{80}$$

**Lemma 14.** *Recall the definition of the projected average reward value function $v_\phi^\pi$ in Equation equation 36. Value function $v_\phi^\pi$ is $2C_m^2 C_p \kappa_r + 2C_m C_r$-Lipschitz in $\Pi$, that is*

$$\left| \left\langle \frac{\partial v_\phi^\pi}{\partial \pi}, \pi' - \pi \right\rangle \right| \le 2\left(C_m^2 C_p \kappa_r + C_m C_r\right) \|\pi' - \pi\|_2, \qquad \forall \pi, \pi' \in \Pi. \tag{81}$$

*Proof.*

$$v_\phi^{\pi_\alpha} = M^{\pi_\alpha}\Phi r^{\pi_\alpha} \tag{82}$$

$$\implies \frac{\partial v_\phi^{\pi_\alpha}}{\partial \alpha} = \frac{\partial M^{\pi_\alpha}}{\partial \alpha}\Phi r^{\pi_\alpha} + M^{\pi_\alpha}\Phi\frac{\partial r^{\pi_\alpha}}{\partial \alpha} \tag{83}$$

$$= M^{\pi_\alpha}\frac{\partial \Phi\mathbb{P}^{\pi_\alpha}}{\partial \alpha}M^{\pi_\alpha}\Phi r^{\pi_\alpha} + M^{\pi_\alpha}\Phi\frac{\partial r^{\pi_\alpha}}{\partial \alpha}, \qquad \text{(from Lemma 13),} \tag{84}$$

$$= M^{\pi_\alpha}\Phi\mathbb{P}^u M^{\pi_\alpha}\Phi r^{\pi_\alpha} + M^{\pi_\alpha}\Phi r^u, \qquad \text{(from equation 69 and equation 73).} \tag{85}$$

$$\implies \left\| \frac{\partial v_\phi^{\pi_\alpha}}{\partial \alpha} \right\|_\infty = \|M^{\pi_\alpha}\Phi\mathbb{P}^u M^{\pi_\alpha}\Phi r^{\pi_\alpha} + M^{\pi_\alpha}\Phi r^u\|_\infty \tag{86}$$

$$\le \|M^{\pi_\alpha}\|_\infty\|\Phi\|_\infty\|\mathbb{P}^u\|_\infty\|M^{\pi_\alpha}\|_\infty\|\Phi r^{\pi_\alpha}\|_\infty + \|M^{\pi_\alpha}\|_\infty\|\Phi r^u\|_\infty \tag{87}$$

$$\le 2C_m^2 C_p \kappa_r + 2C_m C_r. \tag{88}$$

The constants $C_m, C_p, C_r$ and $\kappa_r$ are characterized in Table 1 with their respective bounds in Lemma 18. $\square$

We can now build on the previous lemma to prove the smoothness of the average reward value function.

**Lemma 15.** *The value function $v_\phi^\pi$ is $8(C_m^3 C_p^2 \kappa_r + C_m^2 C_p C_r)$-smooth in $\Pi$. That is,*

$$\left\langle \pi' - \pi, \frac{\partial^2 v_\phi^\pi(s)}{\partial \pi}(\pi' - \pi) \right\rangle \le 8\left(C_m^3 C_p^2 \kappa_r + C_m^2 C_p C_r\right) \|\pi' - \pi\|_2^2 \qquad \forall \pi', \pi \in \Pi, s \in \mathcal{S} \tag{89}$$

*Proof.* From Lemma 14, it is true that

$$\frac{\partial v_\phi^{\pi_\alpha}}{\partial \alpha} = M^{\pi_\alpha}\Phi\mathbb{P}^u M^{\pi_\alpha}\Phi r^{\pi_\alpha} + M^{\pi_\alpha}\Phi r^u$$

$$\implies \frac{\partial^2 v_\phi^{\pi_\alpha}}{\partial \alpha^2} = \frac{\partial}{\partial \alpha}\left[ M^{\pi_\alpha}\Phi\mathbb{P}^u M^{\pi_\alpha}\Phi r^{\pi_\alpha} + M^{\pi_\alpha}\Phi r^u \right]$$

$$= \frac{\partial M^{\pi_\alpha}}{\partial \alpha}\Phi\mathbb{P}^u M^{\pi_\alpha}\Phi r^{\pi_\alpha} + M^{\pi_\alpha}\Phi\mathbb{P}^u\frac{\partial M^{\pi_\alpha}}{\partial \alpha}\Phi r^{\pi_\alpha} + M^{\pi_\alpha}\Phi\mathbb{P}^u M^{\pi_\alpha}\Phi\frac{\partial r^{\pi_\alpha}}{\partial \alpha}$$

$$+ \frac{\partial M^{\pi_\alpha}}{\partial \alpha}\Phi r^u$$

$$= M^{\pi_\alpha}\frac{\partial \Phi\mathbb{P}^{\pi_\alpha}}{\partial \alpha}M^{\pi_\alpha}\Phi\mathbb{P}^u M^{\pi_\alpha}\Phi r^{\pi_\alpha} + M^{\pi_\alpha}\Phi\mathbb{P}^u M^{\pi_\alpha}\frac{\partial \Phi\mathbb{P}^{\pi_\alpha}}{\partial \alpha}M^{\pi_\alpha}\Phi r^{\pi_\alpha}$$

$$+ M^{\pi_\alpha}\Phi\mathbb{P}^u M^{\pi_\alpha}\Phi\frac{\partial r^{\pi_\alpha}}{\partial \alpha} + M^{\pi_\alpha}\frac{\partial \Phi\mathbb{P}^{\pi_\alpha}}{\partial \alpha}M^{\pi_\alpha}\Phi r^u, \qquad \text{(from Lemma 13),}$$

$$= M^{\pi_\alpha}\Phi\mathbb{P}^u M^{\pi_\alpha}\Phi\mathbb{P}^u M^{\pi_\alpha}\Phi r^{\pi_\alpha} + M^{\pi_\alpha}\Phi\mathbb{P}^u M^{\pi_\alpha}\Phi\mathbb{P}^u M^{\pi_\alpha}\Phi r^{\pi_\alpha}$$

$$+ M^{\pi_\alpha}\Phi\mathbb{P}^u M^{\pi_\alpha}\Phi r^u + M^{\pi_\alpha}\Phi\mathbb{P}^u M^{\pi_\alpha}\Phi r^u, \qquad \text{(from equation 69 and equation 73).}$$

Considering the $L_\infty$ norm,

$$
\begin{aligned}
\left\|\frac{\partial^2 v_\phi^{\pi_\alpha}}{\partial \alpha^2}\right\|_\infty &= 2\|M^{\pi_\alpha}\Phi\mathbb{P}^u M^{\pi_\alpha}\Phi\mathbb{P}^u M^{\pi_\alpha}\Phi r^{\pi_\alpha} + M^{\pi_\alpha}\Phi\mathbb{P}^u M^{\pi_\alpha}\Phi r^u\|_\infty \\
&\leq 2\|M^{\pi_\alpha}\Phi\mathbb{P}^u M^{\pi_\alpha}\Phi\mathbb{P}^u M^{\pi_\alpha}\Phi r^{\pi_\alpha}\|_\infty + \|M^{\pi_\alpha}\Phi\mathbb{P}^u M^{\pi_\alpha}\Phi r^u\|_\infty \\
&\leq 8(C_m^3 C_p^2 \kappa_r + C_m^2 C_p C_r).
\end{aligned}
$$

Hence, we obtain,

$$
\left\langle \pi' - \pi, \frac{\partial^2 v_\phi^\pi(s)}{\partial \pi}(\pi' - \pi) \right\rangle \leq 8 \left( C_m^3 C_p^2 \kappa_r + C_m^2 C_p C_r \right) \|\pi' - \pi\|_2^2 \qquad \forall \pi', \pi \in \Pi, s \in \mathcal{S} \quad (90)
$$

$\square$

### A.4 Lipschitzness of the Infinite Horizon Average Reward $\rho^\pi$

The Lipschitzness and smoothness of the projected value function $v_\phi^\pi$ is leveraged through the average reward Bellman equation to prove the Lipschitzness and smoothness of the infinite horizon average reward.

**Lemma 16.** *Recall the average reward Bellman Equation corresponding to a policy $\pi$ and projected value function $v_\phi^\pi$ in Equation equation 32. The average reward $\rho^\pi$ is $L_1^\Pi$-Lipschitz.*

$$
\left| \left\langle \frac{\partial \rho^\pi}{\partial \pi}, \pi' - \pi \right\rangle \right| \leq L_1^\Pi \|\pi' - \pi\|_2, \qquad \forall \pi, \pi' \in \Pi, \quad (91)
$$

*where $L_1^\Pi = 2(C_r + C_p C_m \kappa_r + 2(C_m^2 C_p \kappa_r + C_m C_r))$*

*Proof.* From Equation equation 32,

$$
\rho^\pi \mathbb{1} = r^\pi + \mathbb{P}^\pi v_\phi^\pi - v_\phi^\pi. \quad (92)
$$

Taking derivative with respect to $\alpha$,

$$
\frac{\partial \rho^{\pi_\alpha}}{\partial \alpha}\mathbb{1} = \frac{\partial r^{\pi_\alpha}}{\partial \alpha} + \frac{\partial \mathbb{P}^{\pi_\alpha}}{\partial \alpha}v_\phi^{\pi_\alpha} + \mathbb{P}^{\pi_\alpha}\frac{\partial v_\phi^{\pi_\alpha}}{\partial \alpha} - \frac{\partial v_\phi^{\pi_\alpha}}{\partial \alpha} \quad (93)
$$

$$
(94)
$$

$$
= \Phi r^u + \Phi\mathbb{P}^u v^{\pi_\alpha} + (\mathbb{P}^{\pi_\alpha} - I)(M^{\pi_\alpha}\Phi\mathbb{P}^u M^{\pi_\alpha}\Phi r^{\pi_\alpha} + M^{\pi_\alpha}\Phi r^u), \qquad \text{(from Lemma 14)} \quad (95)
$$

$$
= \Phi r^u + \Phi\mathbb{P}^u M^{\pi_\alpha}\Phi r^{\pi_\alpha} + (\mathbb{P}^{\pi_\alpha} - I)(M^{\pi_\alpha}\Phi\mathbb{P}^u M^{\pi_\alpha}\Phi r^{\pi_\alpha} + M^{\pi_\alpha}\Phi r^u), \quad (96)
$$
$$
\text{(from equation 73 and equation 69).} \quad (97)
$$

Considering the $L_\infty$ norm of the above expression,

$$
\left|\frac{\partial \rho^{\pi_\alpha}}{\partial \alpha}\right| = \left\| \Phi r^u + \Phi\mathbb{P}^u M^{\pi_\alpha}\Phi r^{\pi_\alpha} + (\mathbb{P}^{\pi_\alpha} - I)(M^{\pi_\alpha}\Phi\mathbb{P}^u M^{\pi_\alpha}\Phi r^{\pi_\alpha} + M^{\pi_\alpha}\Phi r^u) \right\|_\infty, \quad (98)
$$

$$
\leq \|\Phi r^u\|_\infty + \|\Phi\mathbb{P}^u M^{\pi_\alpha}\Phi r^{\pi_\alpha}\|_\infty + \|\mathbb{P}^{\pi_\alpha} - I\|_\infty(\|M^{\pi_\alpha}\Phi\mathbb{P}^u M^{\pi_\alpha}\Phi r^{\pi_\alpha}\|_\infty + \|M^{\pi_\alpha}\Phi r^u\|_\infty), \quad (99)
$$

$$
\leq 2\|r^u\|_\infty + 2\|\mathbb{P}^u M^{\pi_\alpha}\Phi r^{\pi_\alpha}\|_\infty + \|\mathbb{P}^{\pi_\alpha} - I\|(\|M^{\pi_\alpha}\Phi\mathbb{P}^u M^{\pi_\alpha}\Phi r^{\pi_\alpha}\|_\infty + \|M^{\pi_\alpha}\Phi r^u\|_\infty), \quad (100)
$$

$$
\leq 2C_r + 2C_p C_m \kappa_r + 2(2C_m^2 C_p \kappa_r + 2C_m C_r). \quad (101)
$$

$\square$

## A.5 SMOOTHNESS OF THE INFINITE HORIZON AVERAGE REWARD $\rho^\pi$

**Lemma 17.** *The average reward $\rho^\pi$ is $L_2^\Pi$-smooth.*

$$\left| \left\langle \pi' - \pi, \frac{\partial^2 \rho^\pi}{\partial \pi^2}(\pi' - \pi) \right\rangle \right| \le \frac{L_2^\Pi}{2} \|\pi' - \pi\|_2^2 \qquad \forall \pi, \pi' \in \Pi, \tag{102}$$

*where $L_2^\Pi = 4(C_p^2 C_m^2 \kappa_r + C_p C_m C_r + (C_p + 1)(C_m^2 C_p \kappa_r + C_m C_r) + 4(C_m^3 C_p^2 \kappa_r + C_m^2 C_p C_r))$.*

*Proof.* From Lemma 16, we have

$$\frac{\partial \rho^{\pi_\alpha}}{\partial \alpha} \mathbb{1} = \Phi r^u + \Phi \mathbb{P}^u M^{\pi_\alpha} \Phi r^{\pi_\alpha} + (\mathbb{P}^{\pi_\alpha} - I)(M^{\pi_\alpha} \Phi \mathbb{P}^u M^{\pi_\alpha} \Phi r^{\pi_\alpha} + M^{\pi_\alpha} \Phi r^u). \tag{103}$$

Taking the derivative again, and repeatedly invoking Equations equation 69, equation 73 and Lemma 13, it follows that,

$$\frac{\partial^2 \rho^{\pi_\alpha}}{\partial \alpha^2} \mathbb{1} = 0 + \frac{\partial}{\partial \alpha}(\Phi \mathbb{P}^u M^{\pi_\alpha} \Phi r^{\pi_\alpha}) + \frac{\partial}{\partial \alpha}\left( (\mathbb{P}^{\pi_\alpha} - I)(M^{\pi_\alpha} \Phi \mathbb{P}^u M^{\pi_\alpha} \Phi r^{\pi_\alpha} + M^{\pi_\alpha} \Phi r^u) \right) \tag{104}$$

$$= \Phi \mathbb{P}^u M^{\pi_\alpha} \Phi \mathbb{P}^u M^{\pi_\alpha} \Phi r^{\pi_\alpha} + \Phi \mathbb{P}^u M^{\pi_\alpha} \Phi r^u + (\mathbb{P}^u)(M^{\pi_\alpha} \Phi \mathbb{P}^u M^{\pi_\alpha} \Phi r^{\pi_\alpha} + M^{\pi_\alpha} \Phi r^u) \tag{105}$$

$$+ (\mathbb{P}^{\pi_\alpha} - I)\left( M^{\pi_\alpha} \Phi \mathbb{P}^u M^{\pi_\alpha} \Phi \mathbb{P}^u M^{\pi_\alpha} \Phi r^{\pi_\alpha} + M^{\pi_\alpha} \Phi \mathbb{P}^u M^{\pi_\alpha} \Phi \mathbb{P}^u M^{\pi_\alpha} \Phi r^{\pi_\alpha} \right. \tag{106}$$

$$\left. + M^{\pi_\alpha} \Phi \mathbb{P}^u M^{\pi_\alpha} \Phi r^u + M^{\pi_\alpha} \Phi \mathbb{P}^u M^{\pi_\alpha} \Phi r^u \right), \tag{107}$$

$$= \Phi \mathbb{P}^u M^{\pi_\alpha} \Phi \mathbb{P}^u M^{\pi_\alpha} \Phi r^{\pi_\alpha} + \Phi \mathbb{P}^u M^{\pi_\alpha} \Phi r^u + (\mathbb{P}^u)(M^{\pi_\alpha} \Phi \mathbb{P}^u M^{\pi_\alpha} \Phi r^{\pi_\alpha} + M^{\pi_\alpha} \Phi r^u) \tag{108}$$

$$+ 2(\mathbb{P}^{\pi_\alpha} - I)\left( M^{\pi_\alpha} \Phi \mathbb{P}^u M^{\pi_\alpha} \Phi \mathbb{P}^u M^{\pi_\alpha} \Phi r^{\pi_\alpha} + M^{\pi_\alpha} \Phi \mathbb{P}^u M^{\pi_\alpha} \Phi r^u \right). \tag{109}$$

Considering the $L_\infty$ norm of the above expression,

$$\left| \frac{\partial^2 \rho^{\pi_\alpha}}{\partial \alpha^2} \right| \le \|\Phi \mathbb{P}^u M^{\pi_\alpha} \Phi \mathbb{P}^u M^{\pi_\alpha} \Phi r^{\pi_\alpha}\|_\infty + \|\Phi \mathbb{P}^u M^{\pi_\alpha} \Phi r^u\|_\infty + \|(\mathbb{P}^u)(M^{\pi_\alpha} \Phi \mathbb{P}^u M^{\pi_\alpha} \Phi r^{\pi_\alpha}$$

$$+ M^{\pi_\alpha} \Phi r^u)\|_\infty + 2\|(\mathbb{P}^{\pi_\alpha} - I)\left( M^{\pi_\alpha} \Phi \mathbb{P}^u M^{\pi_\alpha} \Phi \mathbb{P}^u M^{\pi_\alpha} \Phi r^{\pi_\alpha} + M^{\pi_\alpha} \Phi \mathbb{P}^u M^{\pi_\alpha} \Phi r^u \right)\|_\infty$$

$$\le 4(C_p^2 C_m^2 \kappa_r + C_p C_m C_r + (C_p + 1)(C_m^2 C_p \kappa_r + C_m C_r) + 4(C_m^3 C_p^2 \kappa_r + C_m^2 C_p C_r)).$$

$$\square$$

**Remark:** The smoothness and Lipschitz constant analysis of both the average reward value functions and the infinite horizon average reward are constrained to all directions $u$, such that every $u = \pi - \pi'$ can be expressed as a difference of any two policies $\pi, \pi' \in \Pi$. Hence the smoothness and Lipschitz constants derived are restricted to the directions that can be expressed as this difference and hence are referred to as restricted smoothness/Lipschitzness.

## A.6 TABLE OF CONSTANTS CAPTURING MDP COMPLEXITY

We restate the table of constants and their description here for the sake of convenience.

Table 2: Constants capturing the MDP Complexity

| | Definition | Range | Remark |
|---|---|---|---|
| $C_m$ | $\max_{\pi \in \Pi} \|(I - \Phi P^\pi)^{-1}\|$ | $\frac{2C_e|\mathcal{S}|}{1-\lambda}$ | Lowest rate of mixing |
| $C_p$ | $\max_{\pi, \pi' \in \Pi} \frac{\|P^{\pi'} - P^\pi\|}{\|\pi' - \pi\|_2}$ | $[0, \sqrt{A}]$ | Diameter of transition kernel |
| $C_r$ | $\max_{\pi, \pi'} \frac{\|r^{\pi'} - r^\pi\|_\infty}{\|\pi' - \pi\|_2}$ | $[0, \sqrt{A}]$ | Diameter of reward function |
| $\kappa_r$ | $\max_\pi \|\Phi r^\pi\|_\infty$ | $[0, 2]$ | Variance of reward function |
| $\frac{L_1^\Pi}{2}$ | $C_r + C_p C_m \kappa_r + 2(C_m^2 C_p \kappa_r + C_m C_r)$ | $[0, k_1\sqrt{A}C_m^2]$ | Restricted Lipschitz constant |
| $\frac{L_2^\Pi}{4}$ | $C_p^2 C_m^2 \kappa_r + C_p C_m C_r + (C_p+1)(C_m^2 C_p \kappa_r + C_m C_r) + 4(C_m^3 C_p^2 \kappa_r + C_m^2 C_p C_r)$ | $[0, k_2 A C_m^3]$ | Restricted smoothness constant |

**Lemma 18.** *The constants $C_p, C_r, C_m, \kappa_r$ in Table 2 and other operator norms are bounded as below:*

1. $\|\Phi\| := \max_{\|v\|_\infty \leq 1} \|\Phi v\|_\infty \leq 2.$

2. $\|P^\pi\| = \max_{\|v\|_\infty \leq 1} \|P^\pi v\|_\infty \leq \max_{\|v\|_\infty \leq 1} \|v\|_\infty = 1.$

3. $\kappa_r = \max_\pi \|\Phi r^\pi\|_\infty \leq 2$

4. $C_m \leq \frac{2C_e S}{1-\lambda}$

5. $C_p = \max_{u = \frac{\pi' - \pi}{\|\pi' - \pi\|_2}, \pi', \pi \in \Pi} \max_{\|v\|_\infty \leq 1} \|P^u v\|_\infty \leq \sqrt{A}.$

6. $C_r = \max_{u = \frac{\pi' - \pi}{\|\pi' - \pi\|_2}, \pi', \pi \in \Pi} \|R^u\|_\infty \leq \sqrt{A}.$

*Proof.*   1. Consider the projection matrix $\Phi$,

$$\|\Phi\|_\infty = \max_{\|v\|_\infty \leq 1} \|\Phi v\|_\infty \leq \max \|v\|_\infty \tag{110}$$

$$= \max_{s \in \mathcal{S}} \left| v(s) - \frac{\sum_{s \in \mathcal{S}} v(s)}{S} \right| \tag{111}$$

$$\leq \max_{s \in \mathcal{S}} |v(s)| + \left| \frac{\sum_{s \in \mathcal{S}} v(s)}{S} \right| \tag{112}$$

$$\leq 2\|v\|_\infty = 2 \tag{113}$$

2. The operator norm of $\mathbb{P}^\pi$ is bounded as below:

$$\|P^\pi\| = \max_{\|v\|_\infty \leq 1} \|P^\pi v\|_\infty \tag{114}$$

$$\leq \max_{\|v\|_\infty \leq 1} \|v\|_\infty \tag{115}$$

$$\leq 1. \tag{116}$$

Equality is attained by the vector $v = \mathbb{1}$.

3. $\kappa_r$ is bounded as below:

$$\kappa_r = \max_{\pi \in \Pi} \|\Phi r^\pi\|_\infty \tag{117}$$

$$= \max_{\pi \in \Pi} \left\| r^\pi - \frac{\sum_{s \in \mathcal{S}} r^\pi(s)}{|\mathcal{S}|} \mathbb{1} \right\|_\infty \tag{118}$$

$$\leq \max_{\pi \in \Pi} \|r^\pi\|_\infty + \left\| \frac{\sum_{s \in \mathcal{S}} r^\pi(s)}{|\mathcal{S}|} \mathbb{1} \right\|_\infty \tag{119}$$

$$\leq 2 \tag{120}$$

$\kappa_r$, in some sense, captures the variance of the single step reward function across the class of policies. Greater the variation of the $r$ across different actions, greater the value of $\kappa_r$.

4. $C_m$ is the maximum of the operator norm of the matrix $(I - \Phi\mathbb{P}^\pi)^{-1}$ across all policies $\pi \in \Pi$. It is determined as follows:

$$(I - \Phi\mathbb{P}^\pi)^{-1} = \sum_{k=0}^\infty (\Phi\mathbb{P}^\pi)^k = \sum_{k=0}^\infty \Phi (\mathbb{P}^\pi)^k, \qquad \text{(from Lemma 12)}, \tag{121}$$

$$\stackrel{(a)}{=} \sum_{k=0}^\infty \Phi((\mathbb{P}^\pi)^k - \mathbb{1}(d^\pi)^\top), \qquad (\text{as } \Phi\mathbb{1}(d^\pi)^\top = 0) \tag{122}$$

Let $v \in \mathbb{R}^{|\mathcal{S}|}$ such that $\|v\|_\infty \leq 1$. Then,

$$\implies \|(I - \Phi\mathbb{P}^\pi)^{-1} v\|_\infty \leq \sum_{k=0}^\infty \|\Phi((\mathbb{P}^\pi)^k - \mathbb{1}(d^\pi)^\top)v\|_\infty \tag{123}$$

$$\leq \sum_{k=0}^\infty \|\Phi\|_\infty \|((\mathbb{P}^\pi)^k - \mathbb{1}(d^\pi)^\top)v\|_\infty \tag{124}$$

$$\leq \sum_{k=0}^\infty \|\Phi\|_\infty |\mathcal{S}| \|((\mathbb{P}^\pi)^k - \mathbb{1}(d^\pi)^\top)\|_\infty \|v\|_\infty \tag{125}$$

$$\stackrel{(b)}{\leq} \sum_{k=0}^\infty 2|\mathcal{S}| C_e \lambda^k \|v\|_\infty, \tag{126}$$

$$= \frac{2C_e|\mathcal{S}|}{1 - \lambda} \tag{127}$$

where $d^\pi$ represents the stationary measure associated with the transition kernel $\mathbb{P}^\pi$, (a) follows from the fact that the projection matrix $\Phi$ projects vectors onto a subspace orthogonal to the subspace spanned by the all ones vector $\mathbb{1}$ and (b) is a consequence of the irreducibility and aperiodicity assumption of the Markov chain induced under all policies. More precisely, for any irreducible and aperiodic stochastic matrix $A$, it is true that:

$$\|A^n - \mathbb{1}d^\top\|_\infty \leq C_e \lambda^n, \tag{128}$$

for some constants $\lambda \in [0, 1), C_e < \infty$, where $d$ is stationary distribution of $A$. $\lambda$ is the coefficient of mixing and captures the rate of geometric mixing of the Markov Chain. Hence, higher the value of $\lambda$, lower the rate of mixing.

5. $C_p$ represents the diameter of the transition kernel as a function of the policy class and can be bound as below.

$$C_p := \max_{u = \frac{\pi' - \pi}{\|\pi' - \pi\|_2}, \pi', \pi \in \Pi} \max_{\|v\|_\infty \leq 1} \|\mathbb{P}^u v\|_\infty, \tag{129}$$

$$= \max_{\pi', \pi \in \Pi, \|v\|_\infty \leq 1} \frac{\|(\mathbb{P}^{(\pi' - \pi)} v\|_\infty}{\|\pi' - \pi\|_2}, \tag{130}$$

$$= \max_{\pi', \pi \in \Pi, \|v\|_\infty \leq 1} \max_{s \in \mathcal{S}} \frac{|(\mathbb{P}^{\pi'} v)(s) - (\mathbb{P}^\pi v)(s)|}{\|\pi' - \pi\|_2} \tag{131}$$

$$\leq \max_{\pi', \pi \in \Pi, \|v\|_\infty \leq 1} \max_{s \in \mathcal{S}} \frac{|(\mathbb{P}^{\pi'} v)(s) - (\mathbb{P}^\pi v)(s)|}{\|\pi'_s - \pi_s\|_2}, \qquad \text{(since } \|\pi'_s - \pi_s\|_2 \leq \|\pi' - \pi\|_2\text{)}, \tag{132}$$

$$= \max_{\pi', \pi \in \Pi, \|v\|_\infty = 1} \max_{s \in \mathcal{S}} \frac{|(\pi'_s)^\top \mathbb{P}_s v - (\pi_s)^\top \mathbb{P}_s v|}{\|\pi'_s - \pi_s\|_2}, \quad \text{(where } \mathbb{P}_s(a, s') = P(s'|s, a), \pi_s(a) := \pi(s, a)\text{)}, \tag{133}$$

$$= \max_{\pi', \pi \in \Pi, \|v\|_\infty \leq 1} \max_{s \in \mathcal{S}} \frac{|(\pi'_s - \pi_s)^\top \mathbb{P}_s v|}{\|\pi'_s - \pi_s\|_2}, \tag{134}$$

$$\leq \max_{\pi', \pi \in \Pi, \|v\|_\infty \leq 1} \max_{s \in \mathcal{S}} \frac{\|\pi'_s - \pi_s\|_1 \|\mathbb{P}_s v\|_\infty}{\|\pi'_s - \pi_s\|_2}, \qquad \text{(from Holder's inequality)} \tag{135}$$

$$= \max_{\pi', \pi \in \Pi} \max_{s \in \mathcal{S}} \frac{\|\pi'_s - \pi_s\|_1}{\|\pi'_s - \pi_s\|_2}, \tag{136}$$

$$\leq \sqrt{|\mathcal{A}|}. \tag{137}$$

6. $C_r$ represents the diameter of the single step reward function as a function of the policy class and can be bound as below.

$$C_r := \max_{u = \frac{\pi' - \pi}{\|\pi' - \pi\|_2}, \pi', \pi \in \Pi} \|r^u\|_\infty \tag{138}$$

$$= \max_{\pi', \pi \in \Pi} \frac{\|r^{\pi'} - r^\pi\|_\infty}{\|\pi' - \pi\|_2} \tag{139}$$

$$= \max_{\pi', \pi \in \Pi} \max_{s \in \mathcal{S}} \frac{|r^{\pi'}(s) - r^\pi(s)|}{\|\pi' - \pi\|_2} \tag{140}$$

$$\leq \max_{\pi', \pi \in \Pi} \max_{s \in \mathcal{S}} \frac{|r^{\pi'}(s) - r^\pi(s)|}{\|\pi'_s - \pi_s\|_2}, \qquad \text{(as } \|\pi'_s - \pi_s\|_2 \leq \|\pi' - \pi\|_2\text{)}, \tag{141}$$

$$= \max_{\pi', \pi \in \Pi} \max_{s \in \mathcal{S}} \frac{|(\pi'_s)^\top r_s - (\pi_s)^\top r_s|}{\|\pi'_s - \pi_s\|_2}, \qquad \text{(where } r_s(a) = r(s, a), \pi_s(a) = \pi(s, a)\text{)}, \tag{142}$$

$$C_r = \max_{\pi', \pi \in \Pi} \max_{s \in \mathcal{S}} \frac{|(\pi'_s - \pi_s)^\top r_s|}{\|\pi'_s - \pi_s\|_2}, \tag{143}$$

$$\leq \max_{\pi', \pi \in \Pi} \max_{s \in \mathcal{S}} \frac{\|\pi'_s - \pi_s\|_1 \|r_s\|_\infty}{\|\pi'_s - \pi_s\|_2}, \qquad \text{(from Holder's inequality)} \tag{144}$$

$$= \max_{\pi', \pi \in \Pi} \max_{s \in \mathcal{S}} \frac{\|\pi'_s - \pi_s\|_1}{\|\pi'_s - \pi_s\|_2}, \tag{145}$$

$$\leq \sqrt{|\mathcal{A}|}. \tag{146}$$

Since the directional derivatives considered are all within the policy class, the analysis gives rise to constants such as $C_p$ and $C_r$, which are functions of the underlying policy class. These constants capture the MDP complexity by the virtue of their definition and are an artifact of this proof technique. $\qquad \square$

# B CONVERGENCE OF AVERAGE REWARD PROJECTED POLICY GRADIENT

**Lemma 19.** *For any convex set $\mathcal{X} \subseteq \mathbb{R}^d$, any point $a \in \mathcal{X}$, and any update direction $u \in \mathbb{R}^d$, let $b = \textbf{Proj}_{\mathcal{X}}(a + u)$ be the projection of $a + u$ onto $\mathcal{X}$. It is true that*

  1. $\langle u, b - a \rangle \geq \|b - a\|_2^2$.

  2. $\langle c - b, u - (b - a) \rangle \leq 0, \qquad \forall c \in \mathcal{X}$.

*Proof.* The formal proof can be found in Beck (2014).

However, the proof follows trivially from the geometrical representation of projection (see Figure 3,Kumar et al. (2023)), and the fact that the hyperplane separates a convex set from a point not in the set.

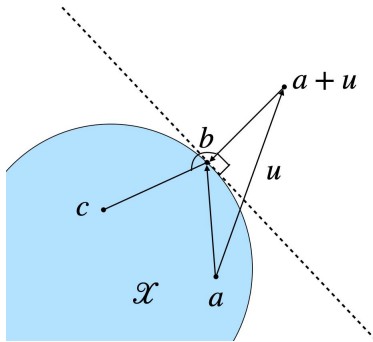

Figure 3: Convex Projection

Intuitively, the proof of the lemma can be interpreted as below.

  1. Since the angle between vectors $(a - b)$ and $((a + u) - b)$ is greater than 90 degrees, it is true that $\langle a - b, (a + u) - b \rangle \leq 0$, which then directly implies $\|b - a\|_2^2 \leq \langle u, b - a \rangle$.

  2. The angle between vectors $(c - b)$ and $((a + u) - b)$ is greater than 90 degrees $\forall c \in \mathcal{X}$, therefore $\langle c - b, u - (b - a) \rangle \leq 0$.

$\square$

## B.1 PROOF OF LEMMA 5

**Lemma 20.** *The average reward iterates $\rho^{\pi_k}$ generated from projected policy gradient satisfy the following,*

$$\rho^{\pi_{k+1}} - \rho^{\pi_k} \geq \frac{L_2^{\Pi}}{2} \|\pi_{k+1} - \pi_k\|^2, \qquad \forall k \geq 0.$$

*where $L_2^{\Pi}$ is the restricted smoothness constant associated with average reward $\rho^{\pi}$.*

*Proof.* From the restricted smoothness of the average cost, we have

$$\rho^{\pi_{k+1}} \geq \rho^{\pi_k} + \left\langle \left. \frac{d\rho^{\pi}}{d\pi} \right|_{\pi=\pi_k}, \pi_{k+1} - \pi_k \right\rangle - \frac{L_2^{\Pi}}{2} \|\pi_{k+1} - \pi_k\|^2,$$

$$= \rho^{\pi_k} + L_2^{\Pi} \left\langle \left. \frac{1}{L_2^{\Pi}} \frac{d\rho^{\pi}}{d\pi} \right|_{\pi=\pi_k}, \pi_{k+1} - \pi_k \right\rangle - \frac{L_2^{\Pi}}{2} \|\pi_{k+1} - \pi_k\|^2,$$

$$\geq \rho^{\pi_k} + L_2^{\Pi} \|\pi_{k+1} - \pi_k\|^2 - \frac{L_2^{\Pi}}{2} \|\pi_{k+1} - \pi_k\|^2.$$

The last inequality follows from the projected gradient ascent policy update rule and item 1 of Lemma 19. Note that the proof only relies on the convexity of the projection set $\Pi$ and the smoothness of the objective function. $\qquad\square$

## B.2 PROOF OF LEMMA 7

**Lemma 21.** *The suboptimality of a policy $\pi$ can be bounded from above as:*

$$\rho^* - \rho^\pi \leq C_{PL} \max_{\pi' \in \Pi} \left\langle \pi' - \pi, \frac{\partial \rho^\pi}{\partial \pi} \right\rangle, \qquad \forall \pi \in \Pi, \tag{147}$$

*where $C_{PL} = \max_{\pi,s} \frac{\partial \pi^*(s)}{\partial \pi(s)}$ and $\rho^*$ is the optimal average reward.*

*Proof.* Average Reward Performance Difference Lemma states that

$$\rho^* - \rho^\pi = \sum_{s \in \mathcal{S}} d^{\pi^*}(s) Q^\pi(s,a) [\pi^*(a|s) - \pi(a|s)] \tag{148}$$

$$\leq \max_{\pi'} \sum_{s \in \mathcal{S}} d^{\pi^*}(s) Q^\pi(s,a) [\pi'(a|s) - \pi(a|s)] \tag{149}$$

$$= \sum_{s \in \mathcal{S}} \frac{d^{\pi^*}(s)}{d^\pi(s)} d^\pi(s) \max_{\pi'_s} Q^\pi(s,a) [\pi'(a|s) - \pi(a|s)] \tag{150}$$

$$= \sum_{s \in \mathcal{S}} \frac{d^{\pi^*}(s)}{d^\pi(s)} \underbrace{d^\pi(s) \max_{\pi'_s} Q^\pi(s,a) [\pi'(a|s) - \pi(a|s)]}_{\geq 0, \quad (= 0 \text{ when } \pi'_s = \pi_s)} \tag{151}$$

$$\leq \sum_{s \in \mathcal{S}} \left( \max_{\pi,s} \frac{d^{\pi^*}(s)}{d^\pi(s)} \right) d^\pi(s) \max_{\pi'_s} Q^\pi(s,a) [\pi'(a|s) - \pi(a|s)] \tag{152}$$

$$= C_{PL} \max_{\pi'} \sum_{s \in \mathcal{S}} d^\pi(s) Q^\pi(s,a) [\pi'(a|s) - \pi(a|s)] \tag{153}$$

$$\overset{(a)}{=} C_{PL} \max_{\pi'} \langle \frac{d\rho^\pi}{d\pi}, \pi' - \pi \rangle, \tag{154}$$

where (a) follows from the average reward policy gradient theorem. $\qquad\square$

## B.3 PROOF OF LEMMA 8

**Lemma 22.** *Let $\pi_{k+1}$ represent the policy iterates obtained through projected policy gradient. For any policy $\pi' \in \Pi$, it is true that,*

$$\left\langle \frac{\partial \rho^{\pi_{k+1}}}{\partial \pi_{k+1}}, \pi' - \pi_{k+1} \right\rangle \leq 4\sqrt{|\mathcal{S}|} L_2^\Pi \|\pi_{k+1} - \pi_k\|_2, \tag{155}$$

*Proof.* For all $x, y \in C$, we have:

$$\left\langle \frac{\partial \rho^{\pi_{k+1}}}{\partial \pi_{k+1}}, \pi' - \pi_{k+1} \right\rangle = \left\langle \frac{\partial \rho^{\pi_{k+1}}}{\partial \pi_{k+1}} - \frac{\partial \rho^{\pi_k}}{\partial \pi_k} + \frac{\partial \rho^{\pi_k}}{\partial \pi_k}, \pi' - \pi_{k+1} \right\rangle \tag{156}$$

$$= \left\langle \frac{\partial \rho^{\pi_{k+1}}}{\partial \pi_{k+1}} - \frac{\partial \rho^{\pi_k}}{\partial \pi_k}, \pi' - \pi_{k+1} \right\rangle + \left\langle \frac{\partial \rho^{\pi_k}}{\partial \pi_k}, \pi' - \pi_{k+1} \right\rangle \tag{157}$$

$$\leq \left\| \frac{\partial \rho^{\pi_{k+1}}}{\partial \pi_{k+1}} - \frac{\partial \rho^{\pi_k}}{\partial \pi_k} \right\| \|\pi' - \pi_{k+1}\| + \left\langle \frac{\partial \rho^{\pi_k}}{\partial \pi_k}, \pi' - \pi_{k+1} \right\rangle \tag{158}$$

$$\overset{(a)}{\leq} L_2^\Pi \|\pi_{k+1} - \pi_k\| \|\pi' - \pi_{k+1}\| + \left\langle \frac{\partial \rho^{\pi_k}}{\partial \pi_k}, \pi' - \pi_{k+1} \right\rangle, \tag{159}$$

where (a) uses smoothness of average reward. Thus, we may continue the chain of inequalities as

$$
equation\ 159 = L_2^\Pi \|\pi_{k+1} - \pi_k\| \|\pi' - \pi_{k+1}\| + \left\langle \frac{\partial \rho^{\pi_k}}{\partial \pi_k} - L_2^\Pi (\pi_{k+1} - \pi_k), \pi' - \pi_{k+1} \right\rangle + L_2^\Pi \left\langle \pi_{k+1} - \pi_k, \pi' - \pi_{k+1} \right\rangle
$$

$$
\leq 2L_2^\Pi \|\pi_{k+1} - \pi_k\| \|\pi' - \pi_{k+1}\| + \left\langle \frac{\partial \rho^{\pi_k}}{\partial \pi_k} - L_2^\Pi (\pi_{k+1} - \pi_k), \pi' - \pi_{k+1} \right\rangle
$$

$$
\leq 2L_2^\Pi \|\pi_{k+1} - \pi_k\| \|\pi' - \pi_{k+1}\| + L_2^\Pi \underbrace{\left\langle \frac{1}{L_2^\Pi} \frac{\partial \rho^{\pi_k}}{\partial \pi_k} - (\pi_{k+1} - \pi_k), \pi' - \pi_{k+1} \right\rangle}_{\leq 0, \qquad \text{(From item 2 of Lemma 19)}}
$$

$$
\leq 2L_2^\Pi \|\pi_{k+1} - \pi_k\| \|\pi' - \pi_{k+1}\|
$$

$$
\leq 2L_2^\Pi \|\pi_{k+1} - \pi_k\| \mathbf{diam}(\Pi).
$$

The diameter of the policy class $\Pi$, can be upper bounded as

$$
\mathbf{diam}(\Pi)^2 = \max_{\pi,\pi} \sum_s \|\pi'_s - \pi_s\|_2^2 \leq \max_{\pi',\pi} \sum_s \|\pi'_s - \pi_s\|_1^2 \leq 4S. \tag{160}
$$

This yields the result. $\qquad\square$

**Lemma 23.** *The scaled sub-optimality $a_k := \rho^* - \rho_k$ follows the recursion*

$$
ca_{k+1}^2 + a_{k+1} - a_k \leq 0, \tag{161}
$$

*where $c = \frac{1}{32 L_2^\Pi |\mathcal{S}| C_{PL}^2}$.*

*Proof.* From Lemma 21, we know that,

$$
\rho^* - \rho^{\pi_{k+1}} \leq C_{PL} \left\langle \pi' - \pi_{k+1}, \frac{\partial \rho^{\pi_{k+1}}}{\partial \pi_{k+1}} \right\rangle, \qquad \forall \pi' \in \Pi, \tag{162}
$$

From Lemma 22, we know that,

$$
\left\langle \frac{\partial \rho^{\pi_{k+1}}}{\partial \pi_{k+1}}, \pi' - \pi_{k+1} \right\rangle \leq 4\sqrt{|\mathcal{S}|} L_2^\Pi \|\pi_{k+1} - \pi_k\|_2. \tag{163}
$$

From Lemma 20, we know that,

$$
\|\pi_{k+1} - \pi_k\|_2 \leq \sqrt{\frac{2\left(\rho^{\pi_{k+1}} - \rho^{\pi_k}\right)}{L_2^\Pi}}, \qquad \forall k \geq 0. \tag{164}
$$

Combining the above equations yields,

$$
\rho^* - \rho^{\pi_{k+1}} \leq \sqrt{32 C_{PL}^2 L_2^\Pi |\mathcal{S}| \left(\rho^{\pi_{k+1}} - \rho^{\pi_k}\right)} \tag{165}
$$

This thus yields,

$$
\frac{(\rho^* - \rho^{\pi_{k+1}})^2}{32 C_{PL}^2 L_2^\Pi |\mathcal{S}|} + (\rho^* - \rho^{\pi_{k+1}}) - (\rho^* - \rho^{\pi_k}) \leq 0. \tag{166}
$$

$\qquad\square$

A more detailed interpretation of this Lemma can be found in Kumar et al. (2023).

### B.4 RECURSION BOUND

In this subsection, we consider the sequence defined as

$$
a_k - a_{k+1} \geq a_k^2,
$$

where $p \geq 0$ and $0 \leq a_0 \leq 1$. Let $f$ be linear interpolation of the sequence $\{a_k\}_{k \geq 0}$, formally defined as

$$
f(x) := (1 - \alpha)a_k + \alpha a_{k+1}, \qquad \text{where } k = \lfloor x \rfloor \text{ and } \alpha = x - \lfloor x \rfloor.
$$

Let $g(0) := a_0$ and

$$
\frac{dg(x)}{dx} = -g(x)^2, \qquad \text{and} \qquad \tau_k := g^{-1}(a_k).
$$

Observe that $g$ is a strictly decreasing function.

**Proposition 1.** *If $\tau_k \geq k$ then*

$$g(x) \geq f(x), \qquad \forall x \in [k, k+1].$$

*Proof.* We have $f(k) = g(\tau_k) = a_k$ and for $\alpha \in (0,1)$

$$\frac{dg(\tau_k + \alpha)}{dx} = -g(\tau_k + \alpha)^2, \qquad \text{(by definition)} \tag{167}$$

$$\geq -g(\tau_k)^2, \qquad (g \text{ is a decreasing function}) \tag{168}$$

$$= -f(k)^2, \qquad (\text{as } f(k) = g(\tau_k)) \tag{169}$$

$$= -a_k^2 \qquad \text{(by definition of } f) \tag{170}$$

$$\geq a_{k+1} - a_k, \qquad \text{(by definition of } a_{k+1}) \tag{171}$$

$$= \frac{df(k+\alpha)}{dx}, \qquad \text{(by definition of } f). \tag{172}$$

Above together with continuity of $f$ and $g$, for all $\alpha \in [0,1]$, we have

$$f(k+\alpha) \leq g(\tau_k + \alpha), \tag{173}$$

$$\leq g(k+\alpha), \qquad (\text{as } \tau_k \geq k \text{ and } g \text{ is a decreasing function}). \tag{174}$$

Hence claim is proved. $\qquad\square$

**Proposition 2.** *For all $k \geq 0$, we have*

$$\tau_k \geq k.$$

*Proof.* Note that $\tau_0 = 0$ by definition $g(0) = a_0$. Now let $\tau_k \geq k$, then from Proposition 1, we have

$$g(x) \geq f(x), \qquad \forall x \in [k, k+1] \tag{175}$$

$$\implies g(k+1) \geq f(k+1) \tag{176}$$

$$= a_{k+1} \tag{177}$$

$$\implies \tau_{k+1} \geq k+1, \qquad (\text{as } g \text{ is a decreasing function}). \tag{178}$$

Hence, by induction the claim is established. $\qquad\square$

**Lemma 24.** *[Recursion Upper Bound] For $p \geq 2$, and $0 \leq a_0 \leq 1$, sequence $\{a_k\}_{k\geq 0}$ satisfying the recursion $a_k - a_{k+1} \geq a_k^2$, follows*

$$a_k \leq \frac{1}{\frac{1}{a_0} + k}, \qquad \forall k \geq 1.$$

*Proof.* From Proposition 2, we get $\tau_k \geq k$. Combining it with Proposition 1, we get

$$g(x) \geq f(x), \qquad \forall x \geq 0.$$

Now, we solve the o.d.e. to get

$$\frac{dg(x)}{dx} = -g(x)^2 \tag{179}$$

$$\implies dx \Big|_{x=0}^{x=k} = \int_{x=0}^{k} -\frac{dg(x)}{g^2(x)} \tag{180}$$

$$\implies k = \frac{1}{g(k)} - \frac{1}{g(0)} \tag{181}$$

$$\implies g(k) = \frac{g(0)}{1 + g(0)k} \tag{182}$$

$$\implies f(k) \leq \frac{a_0}{1 + a_0 k}, \qquad (\text{as } f(x) \leq g(x)) \tag{183}$$

$$\implies a_k \leq \frac{1}{\frac{1}{a_0} + k}. \tag{184}$$

This proves the claim. $\qquad\square$

**Lemma 25.** *If $a_k - a_{k+1} \geq ca_k^2$ then*

$$a_k \leq \frac{1}{\frac{1}{a_0} + ck}.$$

*Proof.* We have

$$a_k - a_{k+1} \geq ca_k^2 \tag{185}$$

$$\implies ca_k - ca_{k+1} \geq (ca_k)^2 \tag{186}$$

$$\implies ca_k \leq \frac{1}{\frac{1}{ca_0} + k}, \qquad \text{(from Lemma 24)} \tag{187}$$

$$\implies a_k \leq \frac{1}{\frac{1}{a_0} + ck}. \tag{188}$$

$\square$

This subsection (proving the recursion upper bound) is inspired by a technique from Kumar et al. (2024). However, while their result is similar, it is not applicable to our case. Their result assumes that $c$ is upper bounded by a constant, which does not hold in our setting. In our case, the smoothness constant (or hardness coefficient) can approach zero, causing the constant $c$ to diverge to infinity. Our result is more general, and the proof technique we use is distinct.

**Lemma 26.** *Given $a_k - a_{k+1} \geq ca_{k+1}^2$, we have*

$$a_k \leq \frac{1}{\frac{1}{a_0} + ck},$$

*where $\nu = c(1 + \frac{8c}{1-\gamma})^{-\frac{3}{2}}$ and $c = \frac{1}{32 C_{PL}^2 |\mathcal{S}| L_2^\Pi}$.*

*Proof.* We have

$$ca_{k+1}^2 + a_{k+1} - a_k \leq 0$$

$$\implies a_{k+1} \leq \frac{-1 + \sqrt{1 + 4ca_k}}{2c},$$

$$= \frac{-1 + f(0) + f'(0)4ca_k + f''(b)\frac{(4ca_k)^2}{2}}{2c}, \qquad \text{(where } f(x) = \sqrt{1+x}, b \in [0, 4ca_k])$$

$$= \frac{2ca_k - 2c^2 a_k^2 (1+b)^{-\frac{3}{2}}}{2c}, \qquad \text{(putting } f(0) = 1, f'(0) = \frac{1}{2}, f''(a) = \frac{(1+b)^{-\frac{3}{2}}}{4})$$

$$= \frac{2ca_k - 2c^2 a_k^2 (1 + 4ca_k)^{-\frac{3}{2}}}{2c}, \qquad \text{(as } b \leq 4ca_k \text{ and } -(1+y)^{-\frac{3}{2}} \text{ is a increasing function )}$$

$$\leq a_k - \frac{ca_k^2}{(1 + 4ca_k)^{\frac{3}{2}}}, \qquad \text{(basic algebra)}$$

$$\leq a_k - \nu a_k^2, \qquad \text{(as } a_k \leq 1 \text{ and } \nu := c(1 + 4c)^{-\frac{3}{2}}).$$

We get the desired result from Lemma 25. $\square$

**Lemma 27.** *If $\frac{1}{c} = 32 C_{PL}^2 |\mathcal{S}| L_2^\Pi < 1$ that is MDP is very easy (i.e. $L_2^\Pi \ll 1$) then the policy gradient converges exponentially fast, that is*

$$a_k \leq \left(\frac{1}{c}\right)^{\frac{k}{2}} a_0^{-2^k}.$$

*Proof.* From the above discussion, we have

$$ca_{k+1}^2 \leq a_k - a_{k+1} \tag{189}$$

$$\leq a_k, \qquad (\text{as } a_{k+1} \geq 0 \text{ by definition}) \tag{190}$$

$$\implies a_{k+1} \leq \sqrt{\frac{a_k}{c}} \tag{191}$$

$$\leq \left(\frac{1}{c}\right)^{\frac{k+1}{2}} a_0^{\frac{1}{2^{k+1}}} \tag{192}$$

$\square$

### B.5 PROOF OF THEOREM 1

We restate the theorem for the sake of convenience.

**Theorem 2.** *Let $\rho^{\pi_k}$ be the average reward corresponding to the policy iterates $\pi_k$, obtained through the policy gradient update equation 6. Let $\rho^*$ represent the optimal average reward, that is, $\rho^* = \max_{\pi \in \Pi} \rho^\pi$. There exist constants $L_2^\Pi$ and $C_{PL}$ which are determined by the underlying MDP such that:*

- *For all MDPs it is true that,*

$$\rho^* - \rho^{\pi_k} \leq \frac{1}{\frac{1}{\rho^* - \rho^{\pi_0}} + \nu k}, \qquad \forall k \geq 0. \tag{193}$$

*where $\nu := \left(\frac{1}{32 C_{PL}^2 |\mathcal{S}| L_2^\Pi}\right) \left(1 + 4\left(\frac{1}{32 C_{PL}^2 |\mathcal{S}| L_2^\Pi}\right)\right)^{-\frac{3}{2}}$*

- *For simple MDPs (i.e. $L_2^\Pi << 1$) we obtain exponential convergence, that is*

$$\rho^* - \rho^{\pi_k} \leq c^{-\frac{k}{2}} \left(\rho^* - \rho^{\pi_0}\right)^{\frac{1}{2^k}}, \qquad \forall k \geq 0 \tag{194}$$

*where $\frac{1}{c} = 32 |\mathcal{S}| L_2^\Pi C_{PL}^2 < 1$.*

*Proof.* Using Gradient Domination Lemma and Sufficient Increase Lemma as shown in Lemma 23, we get the following recursion

$$a_k - a_{k+1} \geq ca_{k+1}^2, \tag{195}$$

where $a_k = J^* - J^{\pi_k}$ and $c = \frac{1}{32 C_{PL}^2 |\mathcal{S}| L_2^\Pi}$ is a small constant. Then we get the desired result by solving the above recursion in Lemma 26 and Lemma 27 for complex MDPs and simple MDPs respectively. $\square$

## C SIMULATION DETAILS

### C.1 CONVERGENCE WITH DIFFERENT ACTION AND STATE SPACE SIZE

In the first experiment, we compare the convergence of PG for a tabular MDP with $(S, A) \in \{(3,3), (9,9), (81,81)\}$. We set the reward kernel to be with maximal variance as described above. For the transition kernel, we use the following matrix:

$$P(\cdot \mid s, \cdot) = \frac{1}{2}\left(1_{S \times A} + \frac{1}{S}\right)$$

so $P(i|s,i) = \frac{1+\frac{1}{S}}{2}$ and $P(i \mid s, j) = \frac{1}{2S}$ for $i \neq j$. For the reward kernel we set the rewards of half the actions to 1 and the rest to $-1$, for every state.

## C.2 Convergence with different reward functions

In the second experiment, we compare the convergence of PG for a tabular MDP with $S = 16$ and $A = 16$. We set $r(s, a) = 0$ for any $s$ and $a$ except for one state which we denote by $s_0$. We use the same randomly generated transition kernel and use the following procedure to generate the reward function:

- No variance: We assign each $(s_0, a)$ pair a reward of 1.
- Low variance: We assign $\frac{1}{8}$ of the actions for $s_0$ a reward of $-1$, and 1 otherwise.
- High variance: We assign $\frac{1}{4}$ of the actions for $s_0$ a reward of $-1$, and 1 otherwise.
- Max variance: We assign $\frac{1}{2}$ of the actions for $s_0$ a reward of $-1$, and 1 otherwise.

## C.3 Convergence with different transition kernels

In the third experiment, we compare the convergence of PG for a tabular MDP with $S = 16$ and $A = 16$. We create three different MDPs with the same $(S, A)$ values and the same reward function that is generated according to the process described above for high variance reward function. We then generate three different transition kernels:

- Uniform: We assign for all values of $s, a, s'$, $P(s' \mid s, a) = \frac{1}{S}$.
- Non-uniform: We assign $P(i \mid s, i) = \frac{1}{2S} + \frac{1}{2}$ and $P(i \mid s, j) = \frac{1}{2S}$ for $i \neq j$.
- Deterministic: We look at $P(\cdot \mid s, \cdot)$ as an $S \times A$ matrix, and assign it a random permutation of the identity matrix. In this way the result MDP is deterministic but not trivial (so every state leads to a different one).

# D Additional Discussion and Future Work

**Extension to Discounted Reward Setting.** In the discounted reward setting, the return $\rho^\pi$, and the value function $v^\pi$ is defined as

$$\rho^\pi = \mu^T (I - \gamma P^\pi) R^\pi, \qquad v^\pi = (I - \gamma P^\pi) R^\pi, \tag{196}$$

where $\gamma \in [0, 1)$ is the discount factor Sutton & Barto (2018). The return $\rho^\pi$ is proven to be $\frac{8}{(1-\gamma)^3}$-smooth (Agarwal et al., 2020). Note that it is an MDP-agnostic bound. We can achieve an MDP instance-dependent bound with a very minor change in the smoothness analysis of the average-reward case.

Let us define,

$$M^{\pi_\alpha} := (I - \gamma P^{\pi_\alpha})^{-1}. \tag{197}$$

Observe that Lemma 13 holds for $A(\alpha) = \gamma P^{\pi_\alpha}$, which yields us

$$\frac{\partial^2 M(\alpha)}{\partial \alpha^2} = \frac{\partial M(\alpha)}{\partial \alpha} \frac{\partial A(\alpha)}{\partial \alpha} M(\alpha) + M(\alpha) \frac{\partial^2 A(\alpha)}{\partial \alpha^2} M(\alpha) + M(\alpha) \frac{\partial A(\alpha)}{\partial \alpha} \frac{\partial M(\alpha)}{\partial \alpha}, \tag{198}$$

where $M(\alpha)$ is shorthand for $M^{\pi_\alpha}$.

Table 3: Constants capturing the MDP Complexity for Discounted reward

| | Definition | Range | Remark |
|---|---|---|---|
| $C_m$ | $\max_{\pi \in \Pi} \|(I - \gamma P^\pi)^{-1}\|$ | $\frac{1}{1-\gamma}$ | |
| $C_p$ | $\gamma \max_{\pi, \pi' \in \Pi} \frac{\|P^{\pi'} - P^\pi\|}{\|\pi' - \pi\|_2}$ | $[0, \gamma\sqrt{A}]$ | Diameter of transition kernel |
| $C_r$ | $\max_{\pi, \pi'} \frac{\|r^{\pi'} - r^\pi\|_\infty}{\|\pi' - \pi\|_2}$ | $[0, \sqrt{A}]$ | Diameter of reward function |
| $\kappa_r$ | $\max_\pi \|r^\pi\|_\infty$ | $[0, 1]$ | Variance of reward function |

**Lemma 28.** *The value function $v_\phi^\pi$ is $8(C_m^3 C_p^2 \kappa_r + C_m^2 C_p C_r)$-smooth in $\Pi$. That is,*

$$\left\langle \pi' - \pi, \frac{\partial^2 v_\phi^\pi(s)}{\partial \pi}(\pi' - \pi) \right\rangle \leq 8\left(C_m^3 C_p^2 \kappa_r + C_m^2 C_p C_r\right) \|\pi' - \pi\|_2^2 \qquad \forall \pi', \pi \in \Pi, s \in \mathcal{S} \quad (199)$$

*Proof.*

$$
\begin{aligned}
\frac{\partial^2 v^{\pi_\alpha}}{\partial \alpha^2} =& M^{\pi_\alpha} \gamma \mathbb{P}^u M^{\pi_\alpha} \gamma \mathbb{P}^u M^{\pi_\alpha} r^{\pi_\alpha} + M^{\pi_\alpha} \gamma \mathbb{P}^u M^{\pi_\alpha} \gamma \mathbb{P}^u M^{\pi_\alpha} r^{\pi_\alpha} \\
&+ M^{\pi_\alpha} \gamma \mathbb{P}^u M^{\pi_\alpha} r^u + M^{\pi_\alpha} \gamma \mathbb{P}^u M^{\pi_\alpha} r^u, \qquad \text{(from equation 69 and equation 73)}.
\end{aligned}
$$

Considering the $L_\infty$ norm,

$$
\begin{aligned}
\left\| \frac{\partial^2 v^{\pi_\alpha}}{\partial \alpha^2} \right\|_\infty &= 2\|M^{\pi_\alpha} \gamma \mathbb{P}^u M^{\pi_\alpha} \gamma \mathbb{P}^u M^{\pi_\alpha} r^{\pi_\alpha} + M^{\pi_\alpha} \gamma \mathbb{P}^u M^{\pi_\alpha} r^u\|_\infty \\
&\leq 2\|M^{\pi_\alpha} \gamma \mathbb{P}^u M^{\pi_\alpha} \gamma \mathbb{P}^u M^{\pi_\alpha} r^{\pi_\alpha}\|_\infty + \|M^{\pi_\alpha} \gamma \mathbb{P}^u M^{\pi_\alpha} r^u\|_\infty \\
&\leq 8(C_m^3 C_p^2 \kappa_r + C_m^2 C_p C_r).
\end{aligned}
$$

Hence, we obtain,

$$\left\langle \pi' - \pi, \frac{\partial^2 v^\pi(s)}{\partial \pi}(\pi' - \pi) \right\rangle \leq 8\left(C_m^3 C_p^2 \kappa_r + C_m^2 C_p C_r\right) \|\pi' - \pi\|_2^2 \qquad \forall \pi', \pi \in \Pi, s \in \mathcal{S} \quad (200)$$

$\square$

The above result implies the return $\rho^\pi$ is $L_2^\Pi = 8(C_m^3 C_p^2 \kappa_r + C_m^2 C_p C_r)$-smooth. This establishes the convergence of the projected policy gradient algorithm with an iteration complexity similar to that stated in Theorem 1 for the average reward case.

**Follow up/concurrent work in discounted reward case:** The work Liu et al. (2024) improved the iteration complexity of the policy gradient (discounted reward case) method to $O(\frac{A}{\epsilon})$ from the previous state-of-the-art iteration complexity of $O(\frac{SA}{\epsilon})$ Xiao (2022a); Mei et al. (2022). Liu et al. (2024) does not use smoothness of the return to establish sufficient increase lemma, instead leverages the performance difference lemma in a novel way.

- Our instance-dependent bound for the MDP stems from the smoothness of the return. As a result, it is unclear how the two approaches can be effectively combined to achieve tighter bounds, leaving this as an avenue for future research.
- However, the bound in Liu et al. (2024) can be asymptotically improved by a factor of $\frac{1}{1-\gamma}$, and more meaningful bounds for initial iterates can be obtained using the enhanced recursion-solving techniques presented in Kumar et al. (2024).

Our work can provide improved, alternative, or suboptimal results (depending on the parameters) for the discounted reward case. However, it remains the first to establish the global convergence of policy gradient methods for the average reward setting.

All existing works (Agarwal et al., 2020; Bhandari & Russo, 2024; Mei et al., 2022; Xiao, 2022a) on the discounted reward case have an iteration complexity that scales with the cardinalities of the state space $S$ and the action space $A$, with the exception of the recent work by Liu et al. (2024), which depends only on $A$. This dependence on the size of the action space poses significant challenges when attempting to generalize to infinite state-action spaces.

**Infinite Action Space.** Our work (for both average and discounted reward case) has the MDP instance bounded bound of $\frac{SL_2^\Pi}{\epsilon}$, where $L_2^\Pi$ is the smoothness constant that encodes the hardness of the MDP. This hardness coefficients that makes up $L_2^\Pi$, may be small/finite for even large/infinite action-space MDPs. However, it requires more careful study to determine the conditions for this to happen, which we leave for the future work.

**Infinite State Space**  For the discounted reward setting, Liu et al. (2024) provides a state-independent bound of $O(\frac{A}{\epsilon})$. The approach taken in their work is fundamentally different from ours, and extending this technique to the average reward case presents an intriguing direction for future research.

In our work, which continues the line of research from Agarwal et al. (2020); Bhandari & Russo (2024); Mei et al. (2022); Xiao (2022a), the bound exhibits state dependence. This dependence arises from the diameter of the policy class, defined as $\text{diam}(\Pi)^2 = \sum_{\pi,\pi'} \|\pi - \pi'\|_2^2 \le S$. While this quantity is inherently tied to the state space, it can potentially be bounded for infinite state spaces under certain structures, such as low-rank policy classes. Exploring this direction in greater detail is an intriguing avenue for future research. Another challenge is to circumvent the dependence on $C_{PL}$ constant which captures the suboptimality of a policy. $C_{PL}$ can be $\infty$ when the state space is infinite, necessitating a different approach to characterizing the suboptimality of a policy.

### D.1 PARAMETRIZED POLICY CLASS LOWER BOUNDS

**Parametrized policy class.**  Our work can also be extended to parameterized policy classes, such as softmax policies. For parameterized policy classes, the smoothness coefficient can be derived by augmenting our analysis with the chain and product rules, which is a straightforward extension. However, this may result in different hardness coefficients, making it an interesting direction for further exploration. We leave this investigation for future work.

**Lower Bounds for policy gradient for average reward case.**  For the discounted reward setting, Mei et al. (2022) establishes a lower bound of $O(\epsilon^{-1})$ for policy gradient methods. Specifically, Theorem 9 of Mei et al. (2022) derives this lower bound using a bandit problem as a counterexample. Since a bandit is a special case of an MDP with a single state, it serves as an example for both discounted reward and average reward MDPs. Consequently, the same lower bound of $O(\epsilon^{-1})$ also applies to average reward MDPs, as implied by Theorem 9 of Mei et al. (2022).

**Linear Rates of Policy Gradient with aggressively increasing step sizes.**  Policy gradient can be interpreted as a form of soft policy iteration, assuming all states are visited or updated by the policy. Specifically, as the learning rate increases, policy gradient behavior increasingly resembles policy iteration. Since policy iteration is known to converge linearly, which is significantly faster than the typical convergence rates of policy gradient methods, it is natural to expect linear convergence bounds for policy gradient with aggressively increasing learning rates. This has been established in several works, including Xiao (2022a); Johnson et al. (2023); Liu et al. (2024).

In most cases, the model is not known, and the exact gradient cannot be computed, requiring the use of stochastic gradient descent. In such noisy settings, using aggressive step sizes can lead to instability in the algorithm.

However, in Theorem 1, we demonstrated linear convergence rates for simple MDPs with constant step sizes. It may also be possible to achieve similar rates for the average reward case by employing aggressively increasing step sizes, which we leave as an interesting direction for future work.

