# OpenReview forum: "Global Convergence of Policy Gradient in Average Reward MDPs"
_ICLR.cc/2025/Conference — ICLR 2025 Poster_

### Official Review · Reviewer_WdYp · 2024-10-31

**Soundness:** 3
**Presentation:** 2
**Contribution:** 2
**Rating:** 6
**Confidence:** 3

**Summary:**

The paper presents the convergence rate analysis of the projected policy gradient algorithm for tabular average reward Markov decision processes (MDPs). Assuming access to the exact gradient, the authors proved a convergence rate of $\mathcal{O}(1/T)$ where $T$ is the number of iterations. To prove the result, they established the smoothness property of the value function for ergodic MDPs, which is of separate interest.

**Strengths:**

1. New state-of-the-art convergence rate of $\mathcal{O}(1/T)$ for projected gradient descent algorithm for average reward MDPs.
2. New smoothness result of the value function for the same setting.
3. Despite some weaknesses stated below, the paper is overall nicely written.

**Weaknesses:**

1. The authors should rewrite the related works and put their work in context. First, they should separate the related works into two groups: ones that use exact gradients (and hence, are more of a planning problem), and others that use gradient estimates (and therefore, are more of a learning problem). Authors should note that some papers explicitly fall into the second group while many others discuss problems of both kinds. The work of the authors falls into the first group. This should be highlighted both in the abstract as well as in the introduction.

2. While mentioning the convergence rate established by earlier works, the authors only focused on the $1-\gamma$ factors while completely ignoring the $\epsilon$ related factor. For example, equation (1) does not show any dependence on $\epsilon$. Is there any specific reason for that? I think it makes the comparison quite confusing.

3. Although one of the results of (Xiao 2022b) proves a convergence rate of $\mathcal{O}\left((1-\gamma)^{-5}\epsilon^{-1}\right)$, in the same paper, they also provide a better result. Specifically, using policy mirror descent, which can be thought of as a generalization of the policy gradient, they establish a linear convergence rate of $\mathcal{O}\left((1-\gamma)^{-1}\log\left((1-\gamma)^{-1}\epsilon^{-1}\right)\right)$. I am surprised that the authors failed to mention the linear convergence rate.

4. Some of the state-of-the-art results mentioned are outdated. For example, (Bai et. al. 2023) is no longer the only work that establishes a regret bound for average reward MDP. A recent paper [1] supersedes their result.

5. To my understanding, the concept of regret makes sense only for a learning problem, not for a planning problem. In my opinion, the author should solely stick to the convergence rate result.

[1] Ganesh, S. and Aggarwal, V., 2024. An accelerated multi-level Monte Carlo approach for average reward reinforcement learning with general policy parametrization. arXiv preprint arXiv:2407.18878.

**Questions:**

1. Since a linear convergence rate is already available in the discounted setup (Xiao 2022b), is it possible to achieve the same in the average reward setup? What are the fundamental challenges to obtain it?

2. Please mention in Table 1 that the constants $C_e$ and $\lambda$ are taken from Assumption 1. It will help the reader.

3. Is the smoothness result only valid for ergodic MDPs or is it possible to extend it to a larger class?

---

> ### Author Response · Authors · 2024-11-21
> **Response to the review**
>
> Thank you for your helpful comments.
>
> **Response to Weaknesses:**
>
> 1. We will rewrite the related work section with these inputs in our revised version.
>
> 2. Since the focus of the paper was to provide the first comprehensive convergence analysis of average reward policy gradient, we focused on why the current state of the art in discounted reward policy gradient could not be leveraged to obtain non trivial bounds in the average reward counterpart. Hence, for sake of clarity we excluded all $\epsilon$ dependences and focused on the role of the discount factor alone in our bounds. However, we will now include the $\epsilon$ dependence in our revised version.
>
> 3. The convergence rate with constant step sizes, is $O(\epsilon^{-1})$ in (Xiao, 2022b), and linear convergence rate of $O(\log\epsilon^{-1})$ with increasing stp sizes.  In our work, we used constant step size, hence we compared our result with its counterpart result in Xiao 2022b. Moreover, we believe that the our result can be extended to attain linear convergence rate with aggressive step sizes as well. But we leave that for future work. However, we note that, while we did not explicitly study the learning problem, increasing step sizes will not work when the $Q$ function has to be estimated and hence, we did not consider the increasing step-size case. However, we have linear convergence in Theorem 1, with constant step sizes for simple MDPs.
>
> 4. Thank you for bringing this paper to our notice. This paper considers the natural policy gradient algorithm whereas we are dealing with projected policy gradient. Besides, they also work under the assumption that the average reward is smooth. Nonetheless, we shall include this paper in our related work section.
>
> 5. We only included regret because such a notion would be useful if one were to extend our result to the case where learning is needed to estimate the policy gradient. For this, we have not discussed regret significantly in the paper. We only mention it in the abstract; we will remove it from the abstract and make a small comment in the paper clarifying when the notion of regret would be useful.
>
> **Response to Questions:**
>
> 1.  (a) The linear rate can be achieved for Projected Mirror Descent (also Policy Gradient) but it requires aggressively increasing step sizes. This aggressive step sizes makes the algorithm very similar to policy iteration (take step sizes to $\infty$). However, this aggressive step sizes are not suitable  when noise is present (or model is not known), and this is where policy gradient shines. Hence, we have limited our study to constant non-aggressive step sizes.
>
>       (b) Moreover, the linear convergence in (Lin Xiao, 2022) requires an additional assumption on the mismatch coefficient. Under this assumption and aggressive step size, it is likely that their analysis applies to our case as well since our sub-optimality recursions are similar.
>
>        (c) We have linear convergence in our Theorem 1 under non-aggressive step sizes for simple MDPs.
>
> 2. We will mention in the table that the definition of $C_e$ and $\lambda$ can be found in Assumption 1.
>
> 3. The proof techniques in this paper depend on the ergodicity of the policy class. Whether non-ergodic MDPs admit a smooth average reward remains an open question. We leave this for future work, anticipating that if true, it will require a fundamentally different proof approach. However, most MDPs can be converted to satisfy our assumption at an $O(\epsilon)$ loss of optimality using the following trick: suppose the original MDP has a probability transition kernel $P(s'|s,a),$ define a new MDP with transition kernel $\frac{\epsilon}{|A|} \sum_{a'}P(s|s,a')+(1-\epsilon)P(s'|s,a),$ where $A$ is the action space. If choosing an action uniformly at random in each state makes the resulting Markov chain irreducible (which is usually the case), then our assumption is automatically satisfied with an $O(\epsilon)$ loss of optimality.

---

### Official Review · Reviewer_D3ZP · 2024-11-03

**Soundness:** 3
**Presentation:** 2
**Contribution:** 2
**Rating:** 6
**Confidence:** 3

**Summary:**

The author show that Project Policy Gradient ascent for average reward MDPs can achieve an $O(\frac{1}{\eps})$ rate to the optimal policy.  To attain this rate, the authors prove the smoothness property of the objective. Additional experiments are conducted to validate the proposed rates.

**Strengths:**

- First proof of global convergence of Project Policy Gradient for average reward MDPs.

**Weaknesses:**

- Missing comparison to [1]. This work improves the convergence rate of [2] and show the rate of Policy Mirror Descentt is linear. Projected Policy Gradient is an instance of Policy Mirror Descent when the squared Eucliden distance is used as the mirror map.
- The clarity of the writing could be improved,
 - The precise definition of $d^\pi(s)$ should be given
 - It's not clear what the step-size used in Thereom 1 Is
- A reference / proof for Eq. 8 should be given.
- Formatting errors: 155: Bellman equation equation 3, 181: discount factorBertsekas (2007), 202: \textit{equation 8}


[1] Johnson, E., Pike-Burke, C., & Rebeschini, P. (2023). Optimal convergence rate for exact policy mirror descent in discounted markov decision processes.
[2] Xiao, L. (2022). On the convergence rates of policy gradient methods.

**Questions:**

- When presenting the convergence rates of the related works, why was the dependence of $\epsilon$ omitted?
- Could the remark of Theorem 1 be clarified. Why is the bound $$\frac{\sigma}{k^p}$$ less meaningful for the inital $k$? Isn't $k$ the number of iterations? Also note that for softmax policies, there exists faster convergence rates shown in [1] compared to [2].
- Is it possible to show that the $O(\frac{1}{\epsilon})$ bound is tight?


[1] Liu, J., Li, W., & Wei, K. (2024). Elementary analysis of policy gradient methods.
[2] Mei, J., Xiao, C., Szepesvari, C., & Schuurmans, D. (2020, November). On the global convergence rates of softmax policy gradient methods.

---

> ### Author Response · Authors · 2024-11-21
> **Response to the review**
>
> Thank you for your helpful comments.
>
> **Response to Weaknesses:**
>
> 1. We thank the reviewer for pointing it out, we will add it this discussion in the final version.
>
>      (a) The work [1, 2] considers discounted reward setting, and our core contribution is in the average reward setting.
>
>      (b) Furthermore, [1,2] achieves a linear rate for PMD with increasing step sizes.  PMD with aggressive step sizes effectively reduces to policy iteration (under suitable condition), which is less suitable for scenarios with noisy gradients (though it is an important yet distinct algorithm).
>
>       (c) We have a linear convergence for simple MDPs with constant step sizes (in average reward case, which can be trivially extended to discounted case.)
>
>       (d) We think, PMD in average case, can have linear convergence too, using the similar techniques in [1,2] and our analysis, with aggressive step sizes. However, it is a different algorithm (with aggressive step sizes PMD becomes closer to PI rather than PG) and hence deserves its own study.
>
> 2. We have fixed typos and improved upon notations in the revised version.
>
> 3. $d^\pi(s)$ is defined and elaborated upon in the paragraph below Equation 2 in the revised version (to be uploaded soon)
>
> 4. The step-size is chosen as $\eta<\frac{1}{L^\Pi_2}$, where $L^\Pi_2$ is the restricted smoothness constant. We will include this in the main theorem statement.
>
> 5. We have added a reference for this result in the revised version.
>
> 6. We have fixed the formatting errors in the revised version.
>
> [1] Johnson, E., Pike-Burke, C., & Rebeschini, P. (2023). Optimal convergence rate for exact policy mirror descent in discounted markov decision processes. [2] Xiao, L. (2022). On the convergence rates of policy gradient methods.
>
> **Response to Questions:**
>
> 1. Since the focus of the paper was to provide the first comprehensive convergence analysis of average reward policy gradient, we focussed on why the current state of the art in discounted reward policy gradient could not be leveraged to obtain non trivial bounds in the average reward counterpart. Hence, for sake of clarity we excluded all $\epsilon$ dependences and focussed on the role of the discount factor in our bounds. However, we will now include the $\epsilon$ dependence in our revised version.
>
> 2.  The existing bounds are of the form $\frac{C}{k(1-\gamma)}$, where $C \gg 1$ is very large constant compared to the worst sub-optimality of $\frac{2}{1-\gamma}$. Now, let's say for $k =10$, we have $\frac{C}{10(1-\gamma)} \gg \frac{2}{1-\gamma}$, which is the largest possible difference in the discounted rewards between two policies. And this is true for all $k \leq C$, which is a very big number. Hence, the bound $\frac{C}{(1-\gamma)k}$, yields a meaningful bound only after a large $k$ which may not be preferable. On the other hand, our bound is of the form $\frac{1}{\frac{1-\gamma}{2} + \frac{k}{C}}$, and this bound is meaningful for all $k\geq 0$. Observe that our bound aysmptotically becomes $\approx \frac{C}{k}$ which improves upon the existing result [2] by a factor $\frac{1}{1-\gamma}$.
> **Regarding softmax policies**: The work [1] improves upon [2], but also uses the recursion $a_k-a_{k-1}\geq a_k^2$ in their work, yielding $\frac{CA}{(1-\gamma)k}$. If our methodology is applied for solving the recursion, it yields the rate of $\frac{1}{\frac{1-\gamma}{2}+\frac{k}{CA}}$. This is a more meaningful bound for small values of $k$ and, asymptotically for large $k,$ provides an improvement by a factor $\frac{1}{1-\gamma}$ asymptotically. We would like to remind, that the work [1,2] is for discounted case, and our core contribution of the work is the analysis on average reward case. The resulting improvement on discounted case is a welcome bonus of our analysis.
>
> 3. This is a good question. A lower bound of $O(\frac{1}{\epsilon})$ is known for policy gradient in the discounted reward case with softmax parametrization. It may be possible to use a similar approach in the average reward case, we will certainly look into this.
>
> [1] Liu, J., Li, W., & Wei, K. (2024). Elementary analysis of policy gradient methods. [2] Mei, J., Xiao, C., Szepesvari, C., & Schuurmans, D. (2020, November). On the global convergence rates of softmax policy gradient methods.

---

> > ### Comment · Reviewer_D3ZP · 2024-11-25
> >
> > Thank you for your response and for the clarifications. I've raised my score to a 6.

---

### Official Review · Reviewer_Z6Q6 · 2024-11-07

**Soundness:** 3
**Presentation:** 2
**Contribution:** 3
**Rating:** 6
**Confidence:** 3

**Summary:**

This paper studies convergence of Policy Gradient (PG) in average-reward MDPs and present non-asymptotic bounds on the global convergence of an error function defined in terms of gains of the optimal policy and the output policy by PG. For the class of unichain MDPs (cf. Assumption 1), the authors present convergence rate to the globally optimal solution (of the reward maximization problem in the long run), but without any assumption on the smoothness of the value functions involved. Such smoothness assumptions were key in the analysis in discounted MDPs. The presented convergence rates decay as $O(1/k)$ where the involved constants depend on MDP-dependent quantities. These results also lead to improved convergence analysis of discounted MDPs.

**Strengths:**

Policy Gradient (PG) and its variants are among interesting and important algorithms in RL. Their convergence properties for the class of discounted MDPs are very well-studied and by now well-understood. However, their counterparts for average-reward MDPs are less explored, especially when the interest lies in globally optimal solution. This is mostly due to the challenges involved in the average-reward setting, rather than the interest in the problem.

One strength of the approach taken in the paper is to depart from the classical approach of using a discounted MDP as a proxy, which further leads to sub-optimal bounds. This way the authors eliminate the smoothness assumption that is typically made in the convergence analysis of PG in the context of discounted MDPs.

The paper admits a good organization. Its technical part is written mostly clearly and precisely, apart from some inconsistent or undefined notations (see comments below). However, there are some inconsistencies in the presentation and advertisement of the results between the introductory part and the main technical part; further on this below. The writing quality is overall fine, but some parts could still benefit from a more careful polishing.

As a positive aspect, the paper delivers a good and accurate review of related literature, to my best knowledge. Yet another positive aspect is reporting numerical results, albeit on toy problems.

**Weaknesses:**

Key Comments and Questions:
-
- The opening of the paper (Abstract and Introduction) talk about regret bounds for PG (scaling as $O(\log(T))$). Figuratively speaking, these are cumulative measures of error incurred by the algorithm. But they are not defined anywhere – or do I miss something? – and the core part of the paper only deals with per-step error measures. Please clarify.
- Despite some interesting results, one key limitation of the paper is the restriction to the class of unichain MDPs (cf. Assumption 1). They are far easier to deal with and are much less relevant in modeling practical RL tasks when compared to the more interesting class of communicating MDPs. Without this assumption, one will not get a closed-form value function in Lemma 1, which is key to establish the results. In other words, it renders unlikely, in my opinion, that the technical tools developed or promoted here could be used beyond the restricted class of MDPs satisfying Assumption 1.
- A key question is how bad the MDP-dependent constant $C_{PL}$ could be. Even though a convergence rate of $O(1/k)$ is superior to those decaying as $O(1/k^p)$ for some $p<1$, the involved MDP-constants (e.g., in Theorem 1) could be prohibitively large in some MDPs (that are not necessarily pathological). More precisely, I expect it could be exponentially large in the size of state-space $|\mathcal S|$.
- In the first paragraph of Section 1, you discuss approaches for determining the optimal policy (i.e., planning algorithms) for average-reward MDPs. Yet you mostly cite papers dealing with the learning problem. Could you clarify, or correct if relevant?

Minor Comments:
-
- In line 50, you use $\pi_k$ but it is not defined yet.
- Regarding refs: Please check formatting guidelines. In many places you must use \citep or \citet instead of \cite so that you get (A & B, year) instead of A & B (year); for instance, in the first paragraph of Section 1. But they are correctly used in Section 1.1. This issue renders rather distracting when reading the paper.
- The work (Lin Xiao, 2022) is cited twice. Is there any difference between them?
- Line 133 (and elsewhere): Using $\Delta(\mathcal A)$ instead of $\Delta \mathcal A$ could make things more readable.
- Inconsistent notations: In Eq. (8) you used $d_\mu(\pi^*)$ whereas later you used $d_{\mu,\gamma}^{\pi^*}$ to denote essentially the same thing.
- Unless I am missing something, the textbook (Boyd and Vandenberghe, 2004) does not include definition of $L$-smoothness, etc.
- Table 1: Make precise the norms used for $C_p$ and $C_m$.

Typos:
-
- Line 82: is , Bai et al. ==> remove “,”
- Line 198: … relationBertsekas …. ==> … relation (Bertsekas, …)
- Line 251: Further is the function is ==> Further if …
- Line 269: euclidean norm ==> Euclidean norm ---- to be consistent with an earlier use of this term.
- Line 346: in the Lemma below ==> … lemma …
- Line 384 and elsewhere in Section 3.2: To be consistent with notations used elsewhere, use $|\mathcal S|$ instead of $S$ since the latter is not defined.
- Line 398: By $L$, did you mean $L_2^{\Pi}$?
- Line 388: a verb might be missing.

**Questions:**

See above.

---

> ### Author Response · Authors · 2024-11-21
> **Response to the review**
>
> Thank you for your helpful comments.
>
> **Response to Key Comments and Questions:**
>
> 1. Theorem 1 states that the suboptimality at iteration $k$ is of $O(\frac{1}{k})$. Hence the total regret accumulated after $T$ iterations of the algorithm is $\sum_{k=1}^T O(\frac{1}{k}) = O(\log T)$. We will include a line on this in the main result.
>
> 2. It is true that Assumption 1 is restrictive when compared to MDP classes such as weakly communicative ones. However, most MDPs can be converted to satisfy our assumption at an $O(\epsilon)$ loss of optimality using the following trick: suppose the original MDP has a probability transition kernel $P(s'|s,a),$ define a new MDP with transition kernel $\frac{\epsilon}{|A|} \sum_{a'}P(s|s,a')+(1-\epsilon)P(s'|s,a),$ where $A$ is the action space. If choosing an action uniformly at random in each state makes the resulting Markov chain irreducible (which is usually the case), then our assumption is automatically satisfied with an $O(\epsilon)$ loss of optimality. Additionally, even if we are able to relax this assumption for the planning problem, often model-free learning requires something like our assumption for algorithms such as TD learning to provide good estimates. This is due to mixing time conditions needed in the analysis of stochastic approximation algorithms (TD learning is an example of one).
>
> 3. We agree with the reviewer that $C_{PL}$ can be quite large. However, it is important to note that $C_{PL}$ is non-trivial in our analysis. In contrast, for discounted reward MDPs, the equivalent constant is $\frac{1}{(1-\gamma)\min_{s}\mu(s)}$, where $\mu(s)$ represents the initial state distribution. As $\gamma$ approaches 1, this constant diverges to infinity, thus providing a vacuous upper bound on the PL constant. In our case, we manage to keep $C_{PL}$ non-trivial, even though it is admittedly large. Moreover, to the best of our knowledge, ours is the first complete proof of the global convergence of policy gradient methods for average-cost problems. Thus, we believe that the fact that $C_{PL}$ is large does not detract from the significance of our contribution; however, we agree with the reviewer that tightening this bound is an important future direction.
>
> 4. In prior average reward literature, most of the work in gradient methods have considered the learning problem rather than planning problem where they make unproven assumptions on the nature of the average reward (such as smoothness). The planning problem without these assumptions remained unsolved but the learning problem has been studied under the assumption that policy gradient converges for the planning problem and hence, we cited those papers. In fact, one of the contributions of our work is in proving smoothness (which was an assumption in previous papers) previously assumed in many learning-based papers.
>
> **Response to Minor Comments:**
>
> 1. $\pi_k$ is the policy obtained at $k$-th iteration of the projected policy gradient algorithm in discounted MDPs. We will include this description in the revised manuscript.
>
> 2. We will make these changes to the citation style in the revised manuscript.
>
> 3. It's the same version. We will remove the redundancy in the revised version.
>
> 4. We will change the notation to $\nabla(\mathcal{A})$ in the revised version.
>
> 5. We will change Eq. 8 to include $d_{\mu,\gamma}^{\pi^*}$ to maintain consistency .
>
> 6. We will remove the citation to Boyd and Vandenberghe.
>
> 7. We have updated the draft, it should be reflected in the final version. $C_p, C_m$ are defined using operator norm w.r.t. $L_\infty$ norm.  Precisely, $C_m = \max_{\pi} \max_{||v||\_\infty\leq 1} ||(I- \Phi P^\pi)\^{-1}v||\_\infty $, and $C_p = \max_{\pi,\pi'\in\Pi}\max_{||v||\_\infty\leq 1}\frac{||(P^{\pi'}- P^\pi)v||\_\infty}{||\pi'-\pi||\_2}$.
>
> **Regarding Typos:**
>
> Thank you for pointing out these typos. We have fixed it in the revised version. Yes, by $L$ we do mean $L_2^\Pi$. We denote $L_2^\Pi$ to specify second derivative continuity on the restricted space $\Pi$.

---

### Official Review · Reviewer_cy7G · 2024-11-10

**Soundness:** 4
**Presentation:** 3
**Contribution:** 3
**Rating:** 8
**Confidence:** 4

**Summary:**

This paper presents a comprehensive global convergence analysis for policy gradient in infinite-horizon average-reward MDPs. It proposes a novel proof framework for the smoothness of the average reward objective, which settles the intrinsic challenge of divergence face by the standard analysis technique that regards the average-reward setting as a limiting case of the discounted-reward setting (as $\gamma \to 1$). Based on the smoothness results, it further analyzes the convergence properties of policy gradient in the average-reward setting, and concludes with an instance-specific bound convergence bound. Simulation results are presented to justify the analysis and reveal the influence of instance-related constants.

**Strengths:**

1. The paper is overall well-written, and the flow is friendly to first-time readers.
2. The research problem is of theoretical interest and importance, which is sufficiently motivated and justified by a thorough review of literature.
3. The technical contributions are solid, rigorous, and clearly articulated (as summarized in Section 1.2). The proofs are checked to be correct and are largely self-contained.
4. Table 1 is especially appreciated since it gives a high-level yet clear idea of the instance-related constants involved in the bound.
5. I like the discussion presented in Section 3.2 that relates the new results to existing results in the classical discounted-reward setting, as well as a brief hint on the reason why instant-specific bounds may be tighter and thus more useful in applications.

**Weaknesses:**

1. The simulation results do help to promote the understanding of the instance-related constants, but it can be improved to include more direct and more convincing evidence under the principle of controlled variables. E.g., exemplary MDP families might be explicitly constructed with certain constant(s) varying and all the others fixed, so that the curves clearly reflect how the performance depends on the varying constant(s).
2. There are a few typesetting issues: (a) Use $\verb|\citep|$ and $\verb|\citet|$ correctly for the author-year format, and avoid using $\verb|\cite|$ — specifically, only use $\verb|\citet|$ when it's a part of the sentence. (b) On line 223 and below, use $\verb|\ll|$ ($\ll$) instead of $<<$. (3) There are a few typos and grammatical issues (e.g., the inconsistency of tenses used in the literature review, where I would recommend the use of present tenses only).

**Questions:**

1. It is briefly touched upon in *Notes on Limitations and Future Work* that the approach can be generalized to "parametric classes of policies". I wonder if the authors have any rough ideas on how this could be done, and further, if it is also doable to extend the tabular MDP setting to generic MDPs with infinite state-action spaces (probably with function approximation, like linear/low-rank MDPs).
2. The relationship with discounted-reward MDPs is discussed in Section 3.2, where it's written that "the constants can be derived through an *analogous* process". Is it possible to (at least) sketch how the final results should look like in the appendix?

---

> ### Author Response · Authors · 2024-11-21
> **Response to the review**
>
> Thank you for your helpful comments.
>
> **Response to Weaknesses:**
>
> 1. Creating MDPs with a fixed $C_p$ is infeasible because modifying the transition kernel inevitably changes $C_m$, preventing the isolation of a single variable's effect. However, for $C_r$, this is precisely what is examined in Figure 1b. In this experiment, all MDPs share the same transition kernel and identical minimal and maximal reward values, ensuring $C_p$, $C_m$, and $\kappa_r$ remain constant. We start by freezing the transition kernel, ensuring that $C_p$ and $C_m$ remain identical across all MDPs. To keep $\kappa_r $ consistent, we fix all rewards to $\pm1$. Then, we vary $C_r$ by adjusting the proportion of actions yielding a reward of $-1$. For the maximal $C_r$, half of the actions return $+1$ and the other half return $-1$. To achieve faster convergence, we increase the proportion of actions that return $ +1$. More details on this can be found in the appendix.
>
> 2. Thank you for the formatting suggestions. We will upload a revised manuscript with these changes soon.
>
> **Response to the Questions:**
>
> 1. This is very interesting direction to explore, thanks for the question.
>
>    A) Parametric classes of policies:  The core idea would be to use the chain rule in our result.
>
>    B) Infinite state-action space:  This is very interesting direction, we thank the reviewer for this insightful question. The traditional bounds are $O(\frac{|S| |A| }{\epsilon})$ which yields vacuous bounds for infinite state-action spaces.
>
>     i) Whereas (for starters), the current version of our result, can deal with infinite action space (possibly with some structure) as our bound is $O(\frac{SL^\Pi_2}{\epsilon})$ and $L^\Pi_2$ depends on the hardness coefficients such as $C_m,C_r,C_p$ which can be finite for infinite action space, hence yielding some meaningful bounds.
>
>     ii) For infinite state space, getting rid of $|S|$ in the bound is challenging. It is obtained from the bound on diameter of the policy class $diam(\Pi)^2 \leq |S|$ (see eq. 160 of the draft). However, it is possible to bound $diam(\Pi)^2$, for policy class with some structure such as low rank policy class for infinite-state space. We thank the reviewer for this intriguing question, we will definitely add this discussion in the main text of the final version. Besides, in order to characterize suboptimality associated with a policy, we  need a finite concentrability coefficient $C_{PL}$ (See Lemma 7). This constant will be infinite when the state space is also infinite. Current approach requires $C_{PL}$ to be finite. Overcoming this state space constraint might require alternate ways to bound the suboptimality.
>
> 2. The extension of this result to the discounted case is as follows. In the smoothness analysis of the discounted MDP case,  $\Phi P^\pi, \Phi R^\pi, $ need to be replaced with the $\gamma P^\pi$ and $R^\pi$ respectively. Only significant change is $C_m = \max_{\pi} \max_{||v||_{\infty} \leq 1} ||(I - \gamma P^{\pi})^{-1} v || \_{\infty} \leq \frac{1}{1-\gamma}$. We will add a subsection in the final version in the appendix, outlining this proof.

---

> > ### Comment · Reviewer_cy7G · 2024-11-25
> > **Thank you for your response!**
> >
> > The rebuttal is clear and convincing. I'll keep my positive attitude towards the acceptance of this paper.

---

### Meta-Review · Area_Chair_9Bz1 · 2024-12-07

**Metareview:**

The paper considers the global convergence behavior of the vanilla policy gradient method for the average reward Markov decision processes (AMDPs), under the tabular setting with finite state and action spaces.
Unlike previous works that directly make smoothness assumptions without verification, this paper provides a complete proof of smoothness properties based on reasonable uniform ergodicity assumptions and then provides an O(1/T) finite-time convergence rate result. Based on the novelty and importance of the result, and the authors have well-addressed all the reviewers' comments, we decide to accept this paper.

**Additional Comments On Reviewer Discussion:**

The reviewers mostly raised some minor comments. The main concerns are about the size of constants and the assumptions on MDP (unichain). The authors have proper cleared these concerns.

---

### Decision · Program_Chairs · 2025-01-22

Accept (Poster)